# VIDEO GENERATION WITH LEARNED ACTION PRIOR

## ABSTRACT

Long-term stochastic video generation remains challenging, especially with moving
cameras. This scenario introduces complex interactions between camera movement
and observed pixels, resulting in intricate spatio-temporal dynamics and partial
observability issues. Current approaches often focus on pixel-level image recon-
struction, neglecting explicit modeling of camera motion dynamics. Our proposed
solution incorporates camera motion or action as an extended part of the observed
image state, employing a multi-modal learning framework to simultaneously model
both image and action. We introduce three models: (i) Video Generation with
Learning Action Prior (VG-LeAP) that treats the image-action pair as an augmented
state generated from a single latent stochastic process and uses variational inference
to learn the image-action latent prior; (ii) Causal-LeAP, which establishes a causal
relationship between action and the observed image frame, and learns a seperate
action prior, conditioned on the observed image states along with the image prior;
and (iii) RAFI, which integrates the augmented image-action state concept with a
conditional flow matching framework, demonstrating that this action-conditioned
image generation concept can be extended to other transformer-based architectures.
Through comprehensive empirical studies on robotic video dataset, RoAM, we
highlight the importance of multi-modal training in addressing partially observable
video generation problems.

## 1 INTRODUCTION

Video prediction is a valuable tool for extracting essential information about the environment, utilized
in various applications such as motion planning algorithms Hafner et al. (2019), and autonomous
navigation and traffic management Claussmann et al. (2020); Bhattacharyya et al. (2018). However,
the complex interactions among different moving objects in a scene present significant challenges for
long-term video prediction Finn et al. (2016); Finn & Levine (2017); Mathieu et al. (2016); Villegas
et al. (2017); Gao et al. (2019b); Villegas et al. (2019); Ebert et al. (2017); Sarkar et al. (2021). Recent
approaches include recurrent deep architectures Srivastava et al. (2015); Oh et al. (2015); Vondrick
et al. (2016); Finn et al. (2016); Mathieu et al. (2016); Villegas et al. (2017); Wichers et al. (2018);
Oprea et al. (2022); Liang et al. (2017); Ebert et al. (2017) and latent variational models Denton &
Fergus (2018); Babaeizadeh et al. (2018); Lee et al. (2018) on human action datasets such as KTH
Schuldt et al. (2004), Human3.6M Ionescu et al. (2014) and robotic datasets such as BAIR Robot
Push Ebert et al. (2017). However, these typically involve static cameras and do not capture the
complexities of moving camera scenarios. Recently visual transformers Dosovitskiy et al. (2021);
Ye & Bilodeau (2022); Gao et al. (2022a), diffusion and flow based models Ho et al. (2022); Mei
& Patel (2023); Davtyan et al. (2023); Harvey et al. (2022); Höppe et al. (2022b) have shown great
promise in generating long-term, high-fidelity predictions.

In scenarios where the camera is moving, video frames are influenced by both the inherent scene
dynamics and the motion of the recording platform. This interplay introduces significant challenges,
particularly in partially observable settings, which are common in domains such as autonomous
vehicles and mobile robotics. Previous works by Villegas et al. (2019); Gao et al. (2019a; 2022b);
Zhong et al. (2024) highlight the complexity of modeling interactions between scene dynamics and
camera motion in partially observable video prediction problem and tried to address it with novel
network architecture designs Gao et al. (2022b); Zhong et al. (2024) or larger latent space Villegas
et al. (2019). Existing datasets such as KITTI Geiger et al. (2013), KITTI-360 Liao et al. (2021),
A2D2 et. al (2020), and Caltech's pedestrian dataset Dollar et al. (2011) emphasize this issue in

outdoor autonomous driving scenarios. For indoor robotics, the RoAM dataset Sarkar et al. (2023) has demonstrated the importance of modeling such interactions by including synchronized image-action pairs, enabling a more comprehensive exploration of partially observable video prediction tasks.

Prior studies on action-conditioned video, introducing Atari reinforcement learning Oh et al. (2015) and Introspective Variational Autoencoders Valencia et al. (2021) incorporated actions as extended video generative model states. However, these approaches assumed the availibility of future actions and learned image priors independent of the camera actions. Similarly, Ma et al. (2022), Finn et al. (2016), Nazari et al. (2022), and Nunes et al. (2020) established video prediction frameworks for object manipulation, predominantly focusing on stationary camera setups with pre-computed manipulator end-effector trajectories.

Recently text token based video diffusion Sohl-Dickstein et al. (2015); Ho et al. (2022) models like AnimateDiff Guo et al. (2023), Videocomposer Wang et al. (2024a), Motionctrl Wang et al. (2024b) and Direct-a-video Yang et al. (2024b) uses textual instructions and in some cases the camera parameters like pan and zoom Yang et al. (2024b) to generate high fidality videos. However these models also assume the avaiability of the desired camera movement beforehand. This assumption may work in controlled environments like stationary robotic manipulators with pre-computed end-effector trajectories, but fails in more dynamic scenarios such as moving cameras in unpredictable environments like busy roads or crowded spaces. In these complex, stochastic settings, an ideal approach requires the ability to learn and predict platform actions based on past and predicted image frames, and vice versa.

In this work, we take a step forward by introducing the two following theoretical frameworks that not only incorporate actions into video prediction but simultaneously predict future actions:

**Conditional Independence**: Under the conditional independence assumption, we model image-action pair as an extended system state and simultaneously predict the next image-action from a shared latent stochastic process. This assumption implies that image and action are independent when the generative latent prior is known. Leveraging this principle, we propose two models: **VG-LeAP**, a variational generative framework, and **RAFI**, built on sparsely conditioned flow matching. By introducing RAFI alongside VG-LeAP, we demonstrate the versatility of incorporating conditional independence into contemporary normalized flow matching Behrmann et al. (2019) frameworks.

**Causal Dependence**: In our causal framework we assume the action is taken after observing the image state and then action leads the system to a new state. Thus images and actions are modeled as causally interlinked nodes, reflecting the real-world scenario where a robot or vehicle takes an action based on the current state and observes the next state as a consequence. This model learns separate latent priors for image and action, with a conditional dependency between them. Following this framework we introduce a new model **Causal-LeAP**, a variational generative frameworks.

All the three proposed models: VG-LeAP, Causal-LeAP and RAFI, not only condition the predicted images on the camera actions, but also model and predict the future camera movement. This aspect has been missing in the video predictive frameworks and paves the way for advancements in autonomous navigation, robotic planning, and beyond.

## 2 PRIOR WORKS

Over the past decade, numerous mathematical frameworks have been proposed to model the current image frame $x_t$ from a sequence of frames $x_{1:T-1}$ from video data of dimension $d = [i_h \times i_w \times 3]$. In their seminal work, Denton & Fergus (2018) introduced the stochastic learned prior model (SVG-lp) . This framework posits that a sequence of image frames from a video is generated from a latent Gaussian distribution. The latent distribution is learned through a variational training and inference paradigm using a set of observed image sequences. The current image frame is predicted as $\tilde{x}_t$ conditioned on the past observed frames $x_{1:t-1}$ and a latent variable $z_t$. Given that at the time of prediction $p(z_t)$ is unknown, it is learnt with a posterior distribution $p_\theta(z_t|x_{1:t}) = \mathcal{N}(\mu_\theta(x_{1:t}), \sigma(x_{1:t}))$ approximated by a recurrent network parameterised by $\theta$. The sampled variable $z_t$ is then used to generate the current image frame $x_t$ conditioned on the past observed frames $x_{1:t-1}$. Denton & Fergus (2018) proposed two methods for learning $p_\theta(z_t|x_{1:t})$: (i) with a fixed Gaussian prior and (ii) with a companion prior model $p_\phi(z_t|x_{1:t-1})$ and minimising the KL divergence loss between the two. This learned prior model has subsequently been utilized in various video generation

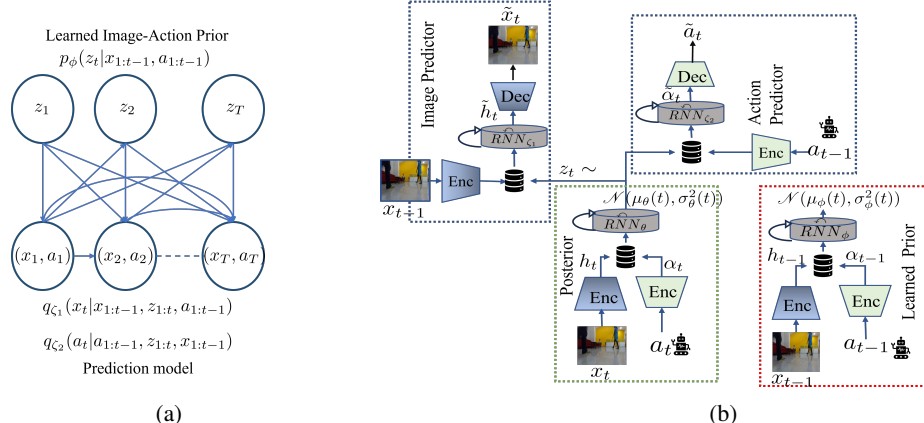

Figure 1: (a) State flow diagram and generation model for VG-LeAP with learned image-action prior $z_t$ dependent on $(x_t, a_t)$. (b) Architecture of video generation with learned action prior (red dotted box) and posterior network (green dotted box). At inference, only the prior model (in red) is used. Prior and posterior latent models are trained using KL divergence loss.

models, such as those by Villegas et al. (2019); Chatterjee et al. (2021) in recent years. However, these frameworks do not address the issue to integrating camera motion with the image generation process in case of action conditioned or moving camera video data.

Camera motion plays a crucial role in the video generation process, especially when the camera is moving or is mounted on a moving platform like a car or a robot. Villegas et al. (2019) showed that with a significantly larger parametric space, SVG-lp can effectively generate and predict future image frames when tested on partially observable video datasets like KITTI, where the camera is mounted on a car. However, recent works, such as those by Sarkar et al. (2023), have demonstrated that long-term video prediction processes can be enhanced by explicitly conditioning the predicted frames on the motion of the camera. Recently, diffusion and flow based models Ho et al. (2022); Davtyan et al. (2023); Voleti et al. (2022); Song et al. (2021); Xu et al. (2020); Höppe et al. (2022a); Guo et al. (2023); Yang et al. (2024b); Wang et al. (2024a) have garnered attention from the computer vision community due to their capacity to generate and forecast high-fidelity video sequences. Rooted in the concepts of diffusion processes Sohl-Dickstein et al. (2015) or Conditional flow matching Lipman et al. (2023), these models iteratively refine noisy data to produce high-quality image frames.

## 3 ACTION CONDITIONED VIDEO GENERATION

We introduce three distinct action-conditioned video generation models. The first two Learned Action Prior or LeAP models: VG-LeAP and Causal-LeAP are variational video generation frameworks in which the image and camera actions are learned through latent Gaussian distributions. However, VG-LeAP is founded on the idea of conditional independence and Causal-LeAP assumes that image and camera actions are linked via causality. With the third model, we introduce RAFI, the Random Action-Frame Conditioned Flow Integrating video generation model, based on RIVER by Davtyan et al. (2023) which uses conditional flow matching. Conditional independence based RAFI shows how camera action conditioning and prediction can be seemlessly integrated into Flow Matching by Lipman et al. (2023) for enhanced video prediction quality.

In this paper, we denote the action of the robot or the platform on which the camera is mounted at timestep $t$ by $a_t \in \mathbb{R}^n$, where $n$ is the dimension of the action or actuation space of the robot/platform. We also assume actions are normalised, that is, $a_t \in [0, 1]$.

### 3.1 VIDEO GENERATION WITH LEARNT ACTION PRIOR (VG-LEAP)

Video generation with Learnt Action Prior, or VG-LeAP, is built on the principles of stochastic video generation in Denton & Fergus (2018). However, unlike Denton & Fergus (2018) where only images were considered as the observed state of the stochastic process, we introduce the notion of

image-action pair $(x_t, a_t)$ as an augmented state of the extended stochastic process that models the image frames as well as the action of the robot. In scenarios where the camera is moving, the observed image frames are influenced by the past actions or movements of the camera. Additionally, in many cases, the future actions of a robotic agent or a car (on which the camera is mounted) depend on the images observed, particularly when obstacle avoidance modules are integrated into the platform's motion planner. This interdependence between the image and action is also referred to as the partial observability problem in video prediction literature Villegas et al. (2017); Sarkar et al. (2021). Thus modelling this process with the notion of system or robot action as a part of an extended state of the process provides a clear way of encapsulating these interdependent dynamics.

We assume that the extended image-action pair $\chi_t = (x_t, a_t)$ is generated from a latent unknown process $p(z_t)$ of variable $z_t$ whose posterior is approximated with a recurrent neural architecture of parameter $\theta$ in the form $p_\theta(z_t|x_{1:t}, a_{1:t})$. In order to learn this posterior distribution, we employ a variational architecture similar to that of SVG-lp. However, in our case, we use the notion of an extended image-action state instead of just the images. We use the reparameterization trick from variational inference Kingma & Welling (2014), to approximate $p_\theta(z_t|x_{1:t}, a_{1:t})$ as a Gaussian process such that $z_t \sim \mathcal{N}(\mu_\theta(z_t|\chi_{1:t}), \sigma_\theta(z_t|\chi_{1:t}))$ where $\mu$ and $\sigma$ denotes the mean and variance. The state flow diagram of the learned image-action prior model in Fig 1a depicts this relationship between learned latent variable $z_t$ and observed image-action pair $(x_t, a_t)$ with connecting blue arrows. We also use a recurrent module parameterised by $\phi$ to learn the image-action prior $p_\phi(z_t|x_{1:t-1}, a_{1:t-1})$ to use during inference when the current image $x_t$ and action $a_t$ are not available. This can also be seen as the learning image-action prior in Fig 1a. The architecture of the network can be expressed as follows and is pictorially represented in Fig 1b:

$$x_t \xrightarrow{Enc} h_t, \quad a_t \xrightarrow{Enc} \alpha_t \quad (1) \qquad \mu_\theta(t), \sigma_\theta(t) = \widehat{RNN}_\theta(h_{0:t}, \alpha_{0:t}), \quad z_t \sim \mathcal{N}(\mu_\theta(t), \sigma_\theta^2(t)) \quad (2)$$

$$x_{t-1} \xrightarrow{Enc} h_{t-1}, \quad \tilde{h}_t = \widehat{RNN}_{\zeta_1}(h_{0:t-1}, z_{1:t}) \tag{3}$$

$$a_{t-1} \xrightarrow{Enc} \alpha_{t-1}, \quad \tilde{\alpha}_t = \widehat{RNN}_{\zeta_2}(\alpha_{0:t-1}, z_{1:t}) \qquad (4) \qquad \tilde{x}_t \xleftarrow{Dec} \tilde{h}_t, \quad \tilde{a}_t \xleftarrow{Dec} \tilde{\alpha}_t \quad (5)$$

In equation 1 we encode image frames to a low dimensional manifold with $h_t$ and map action data to a higher dimensional state of $\alpha_t$. These encoded features are then fed to the posterior estimation network (represented with the green submodule in Fig. 1b) for eventual sampling of $z_t$ in equation 2. Note that the dependence of $z_t$ on past data $(h_{0:t}, \alpha_{0:t})$ arises from the recurrent LSTM components in the posterior network. This same dependence of the predicted image $\tilde{h}_t$ and action data $\tilde{\alpha}_t$ on the history of observed data $(h_{0:t-1}, z_{0:t})$ and $(\alpha_{0:t-1}, z_{0:t})$ in equation 3 and equation 4, are modelled with the LSTM components in the image and action predictor networks $\widehat{RNN}_{\zeta_1}$ and $\widehat{RNN}_{\zeta_2}$. Finally, the generated image $\tilde{x}_t$ and action $\tilde{a}_t$ are decoded with their respective decoder architectures in equation 5. The action conditioned prior $p_\phi(z_t|x_{1:t-1}, a_{1:t-1})$ is learned as $\mu_\phi(t), \sigma_\phi(t) = \widehat{RNN}_\phi(h_{0:t-1}, \alpha_{0:t-1})$ and is shown with the red sub-module in Fig. 1b .

**Loss:** A modified variational lower bound or ELBO loss in equation 6 is used for training.

$$\max_{\theta, \phi, \zeta_1, \zeta_2} \mathcal{L}_{\theta, \phi, \zeta_1, \zeta_2}(x_{1,T}, a_{1:T}) = \sum_{t=1}^{T} \big[ \mathbb{E}_{p_\theta(z_{1:t}|x_{1:t}, a_{1:t})} (\ln q_{\zeta_1}(x_t|x_{1:t-1}, z_{1:t}) +$$

$$\beta_a \ln q_{\zeta_2}(a_t|a_{1:t-1}, z_{1:t})) - \beta D_{KL}(p_\theta(z_t|x_{1:t}, a_{1:t}) || p_\phi(z_t|x_{1:t-1}, a_{1:t-1})) \big] \tag{6}$$

The 1st and 3rd components in equation 6 refer to the widely used reconstruction and KL divergence loss of variational frameworks Denton & Fergus (2018); Villegas et al. (2019); Chatterjee et al. (2021). However, the 2nd term arises from a natural expansion of the extended state of $(x_t, a_t)$ and represents the prediction/reconstruction loss for action $a_t$. In equation 6, $q_{\zeta_1}(x_t|\cdots)$ and $q_{\zeta_2}(a_t|\cdots)$ represents the likelihood functions of predicting $x_t$ and $a_t$, and are estimated with the $L_p$ norm losses (where $p \in \{1, 2\}$) between the ground truth and predicted values. The hyper-parameters $\beta_a$ and $\beta$ are selected based on the numerical stability of training and is discussed in the supplementary material.

### 3.2 CAUSAL VIDEO GENERATION WITH LEARNED ACTION PRIOR

Unlike VG-LeAP, Causal Learned Action Prior or Causal-LeAP does not treat image-action pair $(x_t, a_t)$ as an extended state of a single generative process. Instead, we assume a causal relationship

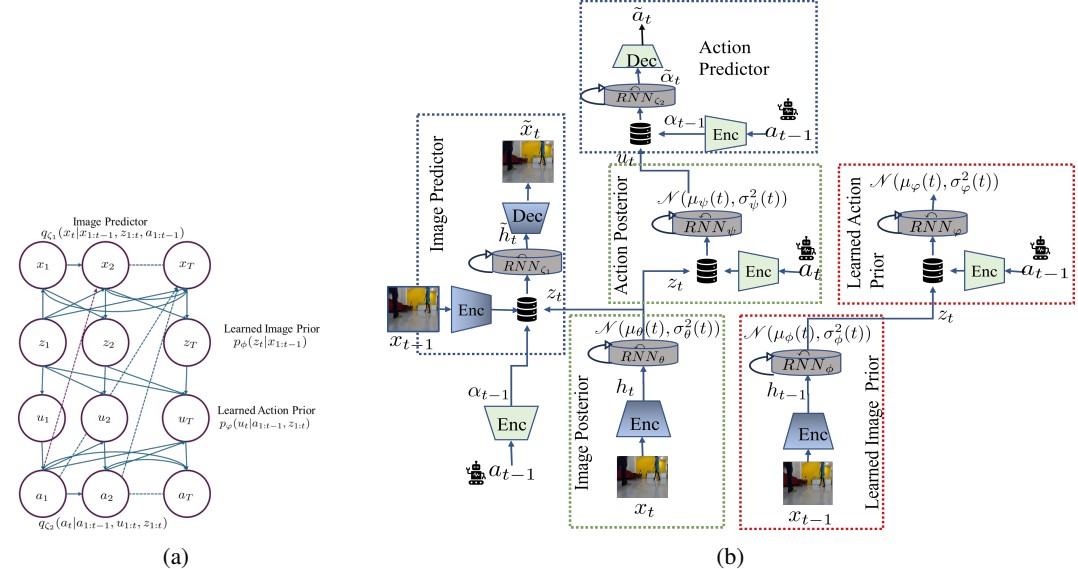

Figure 2: (a) State flow diagram for Causal-LeAP model with learned action prior $u_t$ dependent on image prior $z_t$. Blue line shows forward causal relationship between $z_t$ and $u_t$. Dotted lines from $a_{t-1}$ to $x_t$ show past actions' influence on future images. (b) Architecture with learned action and image prior models (red boxes, used during inference to generate $z_t$ and $u_t$ for $\tilde{x}_t$ and $\tilde{a}_t$). Posterior networks in green boxes.

between the action $a_t$ taken by the moving platform at time-step $t$ and the observed image frame $x_t$. This approach aligns with most motion planning algorithms, where following a Markovian model, action $a_t$ is planned based on the current observed state $x_t$. Consequently, the action taken at time $t$ influences the image frame $x_{t+1}$ observed at $t+1$, and this causal chain continues sequentially with time. Thus, instead of learning a single distribution for both image-action, we learn two different stochastic posteriors: (i) latent image posterior $p_\theta(z_t|x_{1:t})$ that approximates posterior probability of latent variable $z_t$ given our observed images $x_{1:t}$. This is pictorially shown in the upper half portion of the state flow diagram in Fig 2a where $q_{\zeta_1}(x_t|x_{1:t-1}, z_{1:t}, a_{1:t-1})$ represents the probablity of observing $x_t$ given observation history of $x_{1:t-1}, z_{1:t}, a_{1:t-1}$ and (ii) latent action posterior $p_\psi(u_t|a_{1:t}, z_{1:t})$ which approximate the posterior of latent action variable $u_t$ given observations of $a_{1:t}, z_{1:t}$. The causal relationship between image latent variable $z_t$ and action latent variable $u_t$ is shown with blue connecting lines in the lower half portion of the state flow diagram in Fig 2a.

Similar to VG-LeAP, we reparameterize Kingma & Welling (2014), $p_\theta(z_t|x_{1:t})$ and $p_\psi(u_t|a_{1:t}, z_{1:t})$ as Gaussion processes such that $z_t \sim \mathcal{N}(\mu_\theta(z_t|x_{1:t}), \sigma_\theta(x_{1:t}))$ and $u_t \sim \mathcal{N}(\mu_\psi(u_t|a_{1:t}, z_{1:t}), \sigma_\psi(a_{1:t}, z_{1:t}))$, respectively and are represented with the two green sub-modules in the main architecture of Causal-LeAP in Fig 2b. With Causal-LeAP we train two recurrent modules parameterised by $\phi$ and $\varphi$ to learn the image prior $p_\phi(z_t|x_{1:t-1}, a_{1:t-1})$ and causal prior $p_\varphi(u_t|a_{1:t-1}, z_{1:t-1})$ and they are depicted with the two red sub-modules in Fig 2b. $p_\phi(z_t|\cdots)$ and $p_\varphi(u_t|\cdots)$ are used at the time of inference when the current image $x_t$ and action $a_t$ are not available. Comparing Fig. 1b and Fig. 2b, we observe that Causal-LeAP incorporates two additional sub-modules: one for the latent action posterior and another for the latent action prior.

$$x_t \xrightarrow{Enc} h_t, \quad a_t \xrightarrow{Enc} \alpha_t \quad (7) \qquad \mu_\theta(t), \sigma_\theta(t) = \widehat{RNN}_\theta(h_{1:t}), \quad z_t \sim \mathcal{N}(\mu_\theta(t), \sigma_\theta^2(t)) \quad (8)$$

$$\mu_\psi(t), \sigma_\psi(t) = \widehat{RNN}_\psi(\alpha_{1:t}, z_{1:t}), \quad u_t \sim \mathcal{N}(\mu_\psi(t), \sigma_\psi^2(t)) \quad (9)$$

$$x_{t-1} \xrightarrow{Enc} h_{t-1}, \quad \tilde{h}_t = \widehat{RNN}_{\zeta_1}(h_{1:t-1}, z_{1:t}, \alpha_{1:t-1}) \quad (10)$$

$$a_{t-1} \xrightarrow{Enc} \alpha_{t-1}, \quad \tilde{\alpha}_t = \widehat{RNN}_{\zeta_2}(\alpha_{1:t-1}, u_{1:t}) \quad (11) \qquad \tilde{x}_t \xleftarrow{Dec} \tilde{h}_t, \quad \tilde{a}_t \xleftarrow{Dec} \tilde{\alpha}_t \quad (12)$$

Similar to equation 1 of VG-LeAP, we first encode image frames and actions to $h_t$ and $\alpha_t$, in

equation 7 and then feed them to the posterior estimation networks $\widehat{RNN}_\theta$ and $\widehat{RNN}_\psi$ as given in equation 8 and 9. Note that, unlike in equation 2 of VG-LeAP, $z_t$ does not depend upon $a_t$ in equation 8. Equation 9 captures the causal relationship between $x_t$ and $a_t$ as the image latent variable is fed to $\widehat{RNN}_\psi$ to generate $u_t$. The recurrent image and action prediction networks $\widehat{RNN}_{\zeta_1}$ and $\widehat{RNN}_{\zeta_2}$ in equation 10 and equation 11 is similar to equation 3 and equation 4 of VG-LeAP, except that we use the additional action latent variable $u_t$. Finally the generated image $\tilde{x}_t$ and action $\tilde{a}_t$ are decoded with their respective decoder architectures in equation 12. The action conditioned image prior $p_\phi(z_t|\dots)$ is learned as $\mu_\phi(t), \sigma_\phi(t) = \widehat{RNN}_\phi(h_{1:t-1})$ and the causal learned action prior $p_\varphi(u_t|\dots)$ is learned as $\mu_\varphi(t), \sigma_\varphi(t) = \widehat{RNN}_\varphi(\alpha_{1:t-1}, z_{1:t-1})$.

**Loss:** The variational lower bound or ELBO loss, derived below, is used for training.

$$
\begin{aligned}
\max_{\theta,\phi,\psi,\varphi,\zeta_1,\zeta_2} \mathcal{L}_{\theta,\phi,\psi,\varphi,\zeta_1,\zeta_2}(x_{1,T}, a_{1:T}) = \sum_{t=1}^{T} \Big[ & \mathbb{E}_{p_\theta(z_{1:t}|x_{1:t})} \ln q_{\zeta_1}(x_t|x_{1:t-1}, z_{1:t}, a_{1:t-1}) - \\
& \beta D_{KL}(p_\theta(z_t|x_{1:t}) || p_\phi(z_t|x_{1:t-1})) + \beta_a \mathbb{E}_{p_\psi(u_{1:t}|z_{1:t}, a_{1:t})} \ln q_{\zeta_2}(a_t|a_{1:t-1}, u_{1:t}) \\
& - \gamma D_{KL}(p_\psi(u_t|a_{1:t}, z_{1:t}) || p_\varphi(u_t|a_{1:t-1}, z_{1:t})) \Big]
\end{aligned}
\tag{13}
$$

In equation 13, the first two components represent the reconstruction and KL divergence lossses from the likelihood function of the image $x_t$. The third and fourth components come from maximizing the log-likelihood of $p(a_t|x_t)$ or $\ln p(a_t|x_t)$. The third component is the action reconstruction loss and is similar to the second component in equation 6. the fourth component represents the KL divergance between the prior and the posterior distribution over the latent action variable and is a direct consequence of the causal relationship between image and action. The hyper-parameter $\gamma$ relating to the KLD loss associated with the action prior function is chosen according to the numerical stability of the problem. In this case, the action predictor is a much smaller model compared to the image predictor and thus tends to converge much quicker which can lead to numerical instability in case of large learning rates or very small $\beta$ values. The selection criteria for all the three hyper-parameters $\beta, \beta_a$ and $\gamma$ are discussed in the supplementary.

### 3.3 RANDOM ACTION-FRAME CONDITIONED FLOW INTEGRATING VIDEO GENERATOR (RAFI)

The Random Action-Frame Conditioned Flow Integrating video generator or RAFI is based on the sparsely conditioned flow matching model of RIVER by Davtyan et al. (2023). Like RIVER, we also encode our image states in the latent space of a pre-trained VQGAN Esser et al. (2021). However, unlike RIVER, we join the latent image state $z_t$ from the VQGAN network with the action vectors to generate the extended image-action state $\tilde{z}_t$ as shown in the fourth step in Algo. 1. Specifically, $z_t$ has a shape of $[C, H, W]$, where $C$ is the number of channels in the latent space, and $H$ and $W$ are the height and width of the latent representation, respectively. The action vector $a_t$, initially of shape $[A]$ where $A$ is the dimensionality of the action space, is broadcast to $[A, H, W]$ and then concatenated to $z_t$ along the channel dimension. This results in $\tilde{z}_t$ having a shape of $[C + A, H, W]$, effectively integrating action information into every spatial location of the latent representation. Following the creation of $\tilde{z}_t$, we follow steps similar to RIVER to train the flow vector regressor Lipman et al. (2023) using gradient descent. The step-by-step algorithm for RAFI is given in Algo. 1. During inference, after applying the flow-matching process, we obtain $\tilde{z}_t^1$, which maintains the shape of $[C + A, H, W]$. To predict action values, we extract the last $[A, H, W]$ maps from $\tilde{z}_t^1$ and compute their average across the $[H, W]$ spatial dimensions. This operation results in a vector of predicted action values with shape $[A]$, corresponding to the dimensionality of the action space.

## 4 DATASET AND EXPERIMENTS

### 4.1 ROAM DATASET

RoAM or Robot Autonomous Motion dataset is a synchronised and timestamped image-action pair sequence dataset, recorded with a Turtlebot3 Burger robot with a Zed mini stereo camera. The dataset was first introduced by Sarkar et al. (2023) to establish the connection between the generated

---

**Algorithm 1** Training Procedure for RAFI

---

**Require:** Dataset of image, action pair sequence $\mathcal{D}$, number of training iteration $N$
1: **for** $i$ in range$(1, N)$ **do**
2:    Sample a sequence of image frames $x_{1:T}$ and corresponding action sequence $a_{1:T}$ from the dataset $\mathcal{D}$
3:    Encode all the images frames $x_{1:T}$ with a pre-trained VQGAN to obtain $z_{1:T}$
4:    For each $x_t$, concat action $a_t$ as additional channels to the output of VQGAN to get $\tilde{z}_t$
5:    Choose a random target frame $\tilde{z}_\tau, \tau \in \{3, \ldots, T\}$
6:    Sample a timestamp $t \sim U[0, 1]$
7:    Sample a noisy observation $\nu \sim p_t(\tilde{z} \mid \tilde{z}_\tau)$
8:    Calculate target vector filed $\mathcal{U}_t(\nu \mid \tilde{z}_\tau)$
9:    Sample a condition frame $\tilde{z}_c, c \in \{1, \ldots, \tau - 2\}$
10:    Update the parameters $\theta$ of the flow vector field regressor $v_t$ with gradient descent:

$$\nabla_\theta \| v_t(\nu \mid \tilde{z}_{\tau-1}, \tilde{z}_c, \tau - c; \theta) - \mathcal{U}_t(\nu \mid \tilde{z}_\tau) \|^2 \tag{14}$$

11: **end for**

---

image frames and the robot action data. RoAM is recorded indoors capturing corridors, lobby spaces, staircases, and laboratories featuring frequent human movement like walking, sitting down, getting up, standing up, etc. The dataset is segregated into 45 long training video sequences and 5 sequences are kept for testing. The Tensorflow Abadi et al. (2015) Dataset API provided by Sarkar et al. (2023) (comprising more than 300k video sequences, each with 25 frames of image size $64 \times 64 \times 4$) is used to train our frameworks. The dataset also contains the corresponding action values from the robot's motion to capture the movement of the camera. The dimension of the action data in RoAM is $m = 2$ featuring forward velocity along the body $x$-axis and turn rate about the body $z$-axis of the robot's centre of mass and are normalised to values between 0 and 1. More details on the training pipeline are discussed in the experimental setup section of the supplementary.

### 4.2 EXPERIMENTAL SETUP

Out of the 25 frames in each sequence, we randomly select 5 consecutive frames to condition our networks VG-LeAP, Causal-LeAP, SVG, RIVER and RAFI on the past data. All the 5 models generate the next 10 frames in the future during training conditioned on the observed 5 frames. In order to test the networks, we created 1024 randomly generated video sequences of length 40 from the original 5 test sequences in RoAM and tested all the 5 networks against the quantitative performance metrics such as: Peak Signal-to-Noise Ratio (PSNR), VGG16 Cosine Similarity Simonyan & Zisserman (2015), and Fréchet Video Distance (FVD) Unterthiner et al. (2018) and Learned Perceptual Image Patch Similarity or LPIPS metricZhang et al. (2018). Among these metrics, FVD is based on the Fréchet Inception Distance (FID) that is commonly used for evaluating the quality of sequence of images or videos from generative frameworks and measures the similarty between ground truth and the learnt data distributions. We also use VGG16 cosine similarity index, LPIPS and PSNR for frame-wise qunatitative evaluation. The VGG16 cosine similarity index uses the pre-trained VGG16 network Simonyan & Zisserman (2015) to measure the cosine similarity between the generated and ground truth video frames. Recently perceptual similarity metric LPIPS Zhang et al. (2018) which uses pretrained AlexNet as its image feature generator, has emerged as a popular measure Franceschi et al. (2020) for its human-like perception of similarity between two image frames. In case of VGG16 Cosine Similarity and PSNR values, closer resemblance to the ground truth images is indicated by higher values whereas in LPIPS and FVD scores, superior performance is associated with lower values. Each stochastic frameworks is sampled 20 times for each of the 1024 test video snippets.

## 5 RESULTS AND DISCUSSION

During inference, we tested all the proposed models on predicting 20 future frames conditioned on the past 5 image frames and the LPIPS, VGG Cosine Similarity, and PSNR are shown in Fig 3a, 3b and 3c. From all the figures, we can see that Causal-LeAP and VG-LeAP easily outperform SVG-lp on the RoAM dataset. While all these models share similar image predictor architectures, it can be

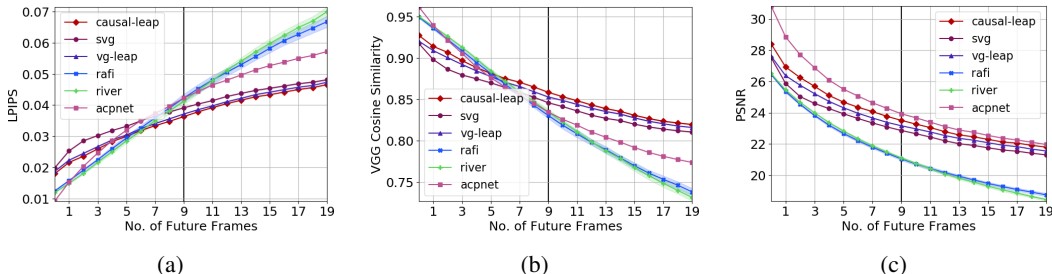

(a)                                  (b)                                  (c)

Figure 3: Quantitative performance comparison of Causal-LeAP, VG-LeAP, SVG (SVG-lp), RAFI, SRVP, and ACPNet for predicting 20 future frames from 5 conditioning frames. (a) LPIPS (lower is better), (b) VGG-16 (higher is better), (c) PSNR (higher is better). Causal-LeAP outperforms others across metrics. RAFI and ACPNet initially outperform Causal-LeAP in LPIPS but decline over time.

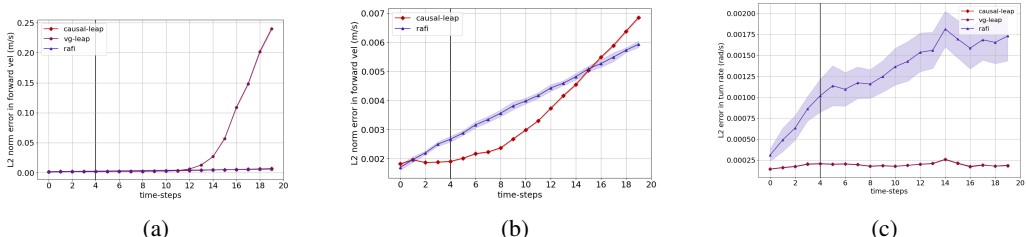

(a)                                  (b)                                  (c)

Figure 4: $L_2$ norm error between predicted and ground truth action values for Causal-LeAP, VG-LeAP, and RAFI. (a) Normalized forward velocity error, with VG-LeAP performing worst. (b) Zoomed view of Causal-LeAP and RAFI velocity errors. (c) Angular rotation/turn rate error, where Causal-LeAP performs best and RAFI worst.

concluded that the improved behaviour is a direct result of modelling the combined image-action dynamics in the case of VG-LeAP and Causal-LeAP. Comparing the behaviour of SVG and VG-LeAP, where both the networks share almost identical architecture and size of the parametric space, VG-LeAP outperforms SVG in Fig. 3a, Fig 3b, and Fig 3c. The mean FVD score of VG-LeAP is around 481.15 which is better than the 539.29 from SVG in Table 1. ACPNet, the only deterministic model in our study, initially generates good predictions (Fig. 3a, 3b) but quickly suffers from blurring effects common in deterministic architectures. ACPNet's FVD score is 908 (Table 1).

Further, Causal-LeAP outperforms VG-LeAP in almost every quantitative metric in Fig. 3a, 3b and 3c, except for FVD score shown in Table 1. Causal-LeAP has an average FVD score of 514.65 compared to 481.15 of VG-LeAP. Both the flow matching based models RIVER and RAFI, initially perform much better than Causal-LeAP and VG-LeAP (Fig 3a,3b), but with time, their performance gets worse. However, in terms of FVD scores, RIVER and RAFI generate the best results with mean scores of 284.46 and 288.23 (Table 1). The poor performace of RIVER and RAFI in terms of PSNR score even after having a good FVD score can be attributed to the fact that PSNR score has a tendency of favouring blurring predictions Zhang et al. (2018); Franceschi et al. (2020) and both the flow matching based frameworks RIVER and RAFI generates very sharp image frames as is in case of any transformer based architectures.

Fig. 4 displays the comparative $L_2$ norm errors for the predicted action data, specifically the normalized forward velocity and turn rate, from Causal-LeAP, VG-LeAP, and RAFI. Figure 4a shows that up to $t = 12$, VG-LeAP, Causal-LeAP, and RAFI produce similar, low $L_2$ norm errors in forward velocity. Beyond $t = 12$, VG-LeAP's error increases exponentially, while Causal-LeAP and RAFI maintain relatively constant errors. This difference stems from VG-LeAP's joint latent variable assumption for the extended image-action state, causing accumulated image errors to adversely affect action predictions. In contrast, Causal-LeAP's separate and causally dependent priors for image and action, enable better long-term action data approximation. If we zoom into Fig. 4a, we can see in Fig. 4b that between RAFI and Causal-LeAP, initially Causal-LeAP performs marginally better that

RAFI, however, after time-step $t = 16$, RAFI provides more accurate forward velocity predictions. However, in case of normalised turn rate, RAFI does not provide reliable predictions as compared to both Causal-LeAP and VG-LeAP shown in Fig 4c.RAFI's erroneous turn rates adversely affect the generated images. This is because RAFI treats image-action as an extended state, causing rotations to result in rotated images, thus decreasing prediction accuracy.

We conducted an ablation study comparing Causal-LEAP, VG-LeAP, and SVG's performance when doubling the frame sampling time-step or $\Delta t_{\text{test}} = 2 \times \Delta t_{\text{train}}$, resulting in videos at 0.5 times the test FPS. This scenario, where people appear to move faster, tests the frameworks' adaptability and generalization. Figures 5a and 5b show the LPIPS and VGG cosine similarity plots for $2 \times \Delta t_{\text{train}}$, respectively. Results indicate that Causal-LeAP outperforms both VG-LeAP and SVG-lp in this modified scenario.

Fig. 6 displays zoomed raw generated frames from Causal-LeAP, VG-LeAP, SVG-lp, and Ground Truth (GT) at selected timestamps, while Fig. 8 shows frames from RAFI, RIVER, and GT. We present the best samples based on VGG cosine similarity from 20 random generations per video sequence. Predicted forward velocities and turn rates from Causal-LeAP and VG-LeAP are shown in Fig. 7a and 7b, corresponding to video sequence in Fig. 6. Fig. 7a demonstrates VG-LeAP's velocity predictions diverging from GT after $t = 18$, while Causal-LeAP maintains accuracy. Fig. 7c and 7d show RAFI's velocity and turn rate predictions for Fig. 8, with Fig. 7c illustrating RAFI's close approximation of GT velocities. Additional raw frame samples are available in the supplementary.

**Discussion:** Our work with RAFI, the action-conditioned flow matching framework, reveals that despite its strong FVD score performance, it struggles with frame-wise reconstruction, as evidenced by the LPIPS and VGG cosine plots in Fig. 3a and 3b. RIVER shows similar poor performance, with RAFI marginally outperforming it in long-term prediction (Fig. 3a). In partially observable scenarios with moving cameras, both conditional flow-based frameworks struggle with long-term prediction. We hypothesize this is due to the problem of crossing conditional paths in conditional flow matching Yang et al. (2024a), where camera movement complicates the network's ability to find diffeomorphic maps. This warrants further investigation in future work.

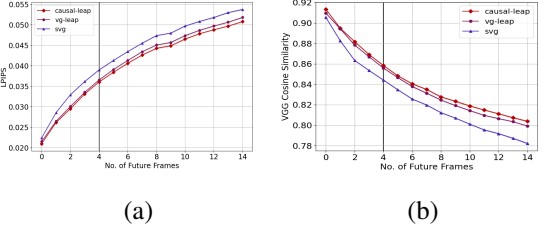

(a)          (b)

| Model | Score |
|---|---|
| Causal-LeAP | $514.65 \pm 3.37$ |
| VG-LeAP | $481.15 \pm 2.39$ |
| SVG-lp | $539.29 \pm 1.94$ |
| **RIVER (BEST)** | **$284.46 \pm 3.21$** |
| **RAFI** | **$288.23 \pm 4.39$** |
| SRVP | $596.68 \pm 2.82$ |
| ACPNET | $908.36$ |

Table 1: FVD Score

Figure 5: A frame-wise ablation study on Causal-LeAP, VG-LeAP and SVG-lp,Fig. 5a and 5b show the LPIPS score and VGG 16 Cosine Similarity respectively, for predicting 15 frames into the future from past 5 frames at 0.5 fps$_{\text{train}}$ or $\Delta t_{\text{test}} = 2 \times \Delta t_{\text{train}}$.

# 6 CONCUSION

We have presented three new stochastic video generative frameworks based on the mathematical premise of incorporating action into the video generation process. We have also established a causal relationship between the image and camera actions in the partially observable scenarios where the camera is moving with our Causal-LeAP model and have shown with our detailed empirical studies that not only image-action models improve the efficacy of the prediction framework but also provides a way to learn and model the system dynamics by simply observing and modelling the interaction between the image-action pair. The causal model learned an action prior conditioned on the latent image state $p_\varphi(u_t | a_{1:t-1}, z_{1:t})$ which can have direct applications to the field of robotics and RL. The model RAFI also shows how easily one can extend the concepts of image-action state pair to existing flow matching approaches leading to useful results and avenues for future research.

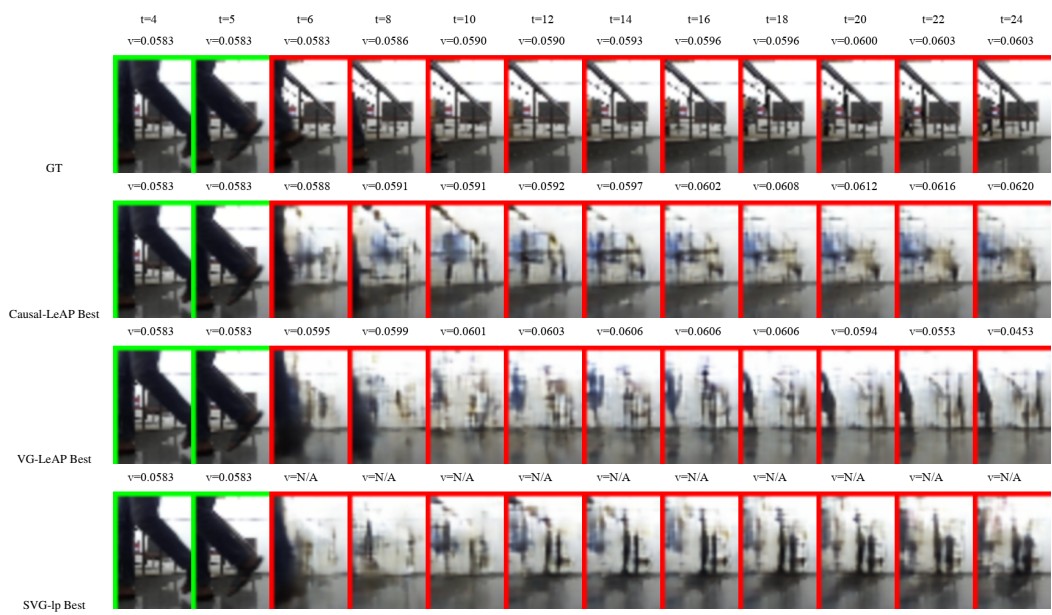

Figure 6: Zoomed Samples (with best VGG cosine similarity) from Causal-LeAP, VG-LeAP and SVG-lp along with Ground Truth. Samples are zoomed with bilinear extrapolation for better visibility. The normalised forward velocities for GT, Causal-LeAP and VG-LeAP are denoted at the top of the frames.

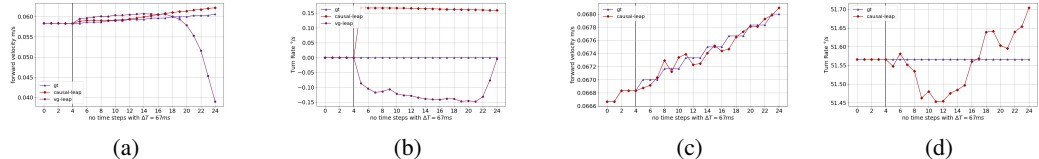

| (a) | (b) | (c) | (d) |

Figure 7: Fig. 7a and 7b shows the predicted forward velocity and turn rates from Causla-LeAP and VG-LeAP along with GT for corresponding video sequence in Fig. 6 and Fig. 7c and 7d shows the predicted forward velocity and turn rates from RAFI along with GT values for Fig. 8

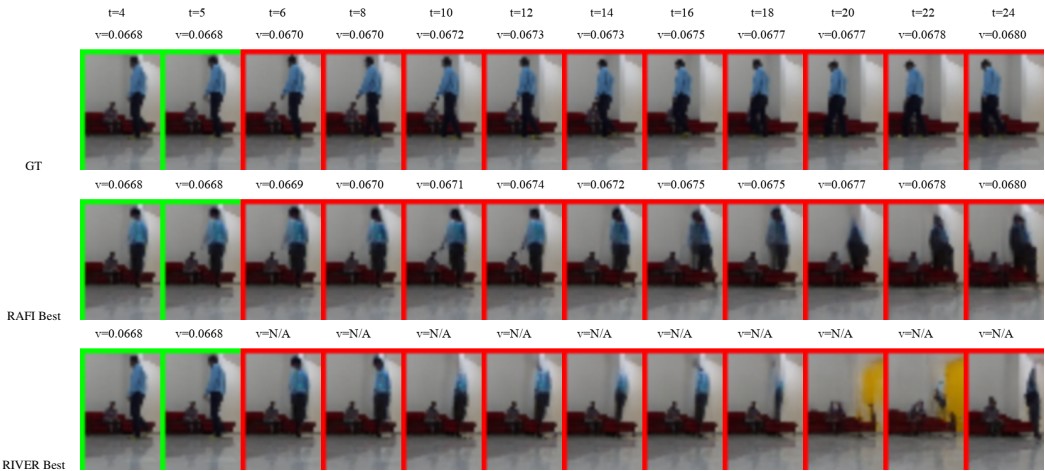

Figure 8: Zoomed Samples (with best VGG cosine similarity) from RAFI and RIVER along with GT. Forward velocities are denoted at the top of the frames.

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
