# Video Generation with Learned Action Prior- Supplementary

## 1 Proof for Video Generation with Learned Action Prior

### 1.1 Variational Lower Bound

Here we are trying to maximize the joint likelihood of $(x_{1:t}, a_{1:T})$ which is equivalent to maximizing $\ln q_\zeta(x_{1:T}, a_{1:T})$ or $\ln q_\zeta(x, a)$ for better readability. Let's assume $z = [z_1, \cdots, z_T]$ denotes the latent $z$ variable across all the time-steps and they are independent of each other across time.

$$\ln q_\zeta(x_{1:T}, a_{1:T}) \equiv \ln q_\zeta(x, a) = \ln \int_z q_\zeta(x, a|z)p(z) \tag{1}$$

$$= \ln \int_z q_\zeta(x, a|z)p(z)\frac{p_\theta(z|x, a)}{p_\theta(z|x, a)} \tag{2}$$

$$= \ln \left( \mathbb{E}_{p_\theta(z|x,a)} q_\zeta(x, a|z)\frac{p(z)}{p_\theta(z|x, a)} \right) \tag{3}$$

$$\geq \mathbb{E}_{p_\theta(z|x,a)} \left( \ln q_\zeta(x, a|z)\frac{p(z)}{p_\theta(z|x, a)} \right) \tag{4}$$

$$= \mathbb{E}_{p_\theta(z|x,a)} \ln q_\zeta(x, a|z) - \mathbb{E}_{p_\theta(z|x,a)} \left( \ln \frac{p_\theta(z|x, a)}{p(z)} \right) \tag{5}$$

$$= \mathbb{E}_{p_\theta(z|x,a)} \ln q_\zeta(x, a|z) - D_{KL}(p_\theta(z|x, a)||p(z)) \tag{6}$$

Given that we assumed in our model that $x$ and $a$ and conditionally independent given $z$, thus $q_\zeta(x, a|z) = q_{\zeta_1}(x|z)q_{\zeta_2}(a|z)$, where $\zeta = \{\zeta_1, \zeta_2\}$. Thus equation 6 can be written as :

$$\ln q_\zeta(x, a) \geq \mathbb{E}_{p_\theta(z|x,a)} \ln q_{\zeta_1}(x|z) + \mathbb{E}_{p_\theta(z|x,a)} \ln q_{\zeta_2}(a|z) - D_{KL}(p_\theta(z|x, a)||p(z)) \tag{7}$$

Similar to SVG, we use RNN architectures in VG-LeAP to recursively predict image frames and at each time-step $t$, $\stackrel{\frown}{RNN}_{\zeta_1}$ takes the encoded past image $x_{t-1}$ and $z_t$ as input. With the recursive behaviour of $\stackrel{\frown}{RNN}_{\zeta_1}$, we can express $\ln q_{\zeta_1}(x|z)$ as:

$$\ln q_{\zeta_1}(x_{1:T}|z_{1:T}) \equiv \ln q_{\zeta_1}(x|z) = \ln \prod_t q_{\zeta_1}(x_t|x_{1:t-1}, z_{1:T}) \tag{8}$$

$$= \sum_t \ln q_{\zeta_1}(x_t|x_{1:t-1}, z_{1:t}) \tag{9}$$

In the case of action predictor in VG-LeAP, we use a similar RNN architecture $\stackrel{\frown}{RNN}_{\zeta_2}$ which takes the past action $a_{t-1}$ and $z_t$ as input. Thus $ln q_{\zeta_2}(a|z)$ can be expressed as:

$$\ln q_{\zeta_2}(a_{1:T}|z_{1:T}) \equiv \ln q_{\zeta_2}(a|z) = \ln \prod_t q_{\zeta_2}(a_t|a_{1:t-1}, z_{1:T}) \tag{10}$$

$$= \sum_t \ln q_{\zeta_2}(a_t|a_{1:t-1}, z_{1:t}) \tag{11}$$

In the case of the posterior and learned prior networks $\stackrel{\frown}{RNN}_\theta$ and $\stackrel{\frown}{RNN}_\phi$ respectively, we recursively feed the action $a_t$ and image $x_t$ to approximate $z_t$ (in case of learned prior we feed $x_{t-1}$ and $a_{t-1}$). Cause $z_t$s are independent across time, $p_\theta(z|x, a)$ can be expressed as:

$$p_\theta(z_{1:T}|x_{1:T}, a_{1:T}) \equiv p_\theta(z|x, a) = \prod_t p_\theta(z_t|x_t, a_t) \tag{12}$$

Submitted to the 38th Conference on Neural Information Processing Systems (NeurIPS 2024) Track on Datasets and Benchmarks. Do not distribute.

We assume the extended image-action state as $\chi = (x, a)$ for better readability and compact expressions in long equations. Since $z_t$s are independent across time, we can rewrite $D_{KL}(p_\theta(z|x, a)||p(z))$ or $D_{KL}(p_\theta(z|\chi)||p(z))$ as:

$$D_{KL}(p_\theta(z|x,a)||p(z)) \equiv D_{KL}(p_\theta(z_{1:T}|x_{1:T}, a_{1:T})||p(z_{1:T})) = \int_z p_\theta(z|x,a)\ln\frac{p_\theta(z|\chi)}{p(z)} \quad (13)$$

$$= \int_{z_1} \cdots \int_{z_T} p_\theta(z_1|\chi_1)\cdots p_\theta(z_T|\chi_{1:T})\ln\frac{p_\theta(z_1|\chi_1)\cdots p_\theta(z_T|\chi_{1:T})}{p(z_1)\cdots p(z_T)} \quad (14)$$

$$= \int_{z_1} \cdots \int_{z_T} p_\theta(z_1|\chi_1)\cdots p_\theta(z_T|\chi_{1:T})\sum_t \ln\frac{p_\theta(z_t|\chi_{1:t})}{p(z_t)} \quad (15)$$

$$= \sum_t \int_{z_1} \cdots \int_{z_T} p_\theta(z_1|\chi_1)\cdots p_\theta(z_T|\chi_{1:T})\ln\frac{p_\theta(z_t|\chi_{1:t})}{p(z_t)} \quad (16)$$

Since $\int_z p_\theta(z) = 1$ we can further simplify equation 16 as

$$D_{KL}(p_\theta(z|x,a)||p(z)) = \sum_t \int_{z_t} p_\theta(z_t|\chi_{1:t})\ln\frac{p_\theta(z_t|\chi_{1:t})}{p(z_t)} \quad (17)$$

$$= \sum_t D_{KL}(p_\theta(z_t|\chi_{1:t})||p(z_t)) = \sum_t D_{KL}(p_\theta(z_t|x_{1:t}, a_{1:t})||p(z_t)) \quad (18)$$

Thus combining equation 9, equation 11and equation 18, with equation 7 we get the variational lower bound as:

$$\ln q_\zeta(x, a) \geq \mathbb{E}_{p_\theta(z|x,a)}\ln q_{\zeta_1}(x|z) + \mathbb{E}_{p_\theta(z|x,a)}\ln q_{\zeta_2}(a|z) - D_{KL}(p_\theta(z|x,a)||p(z)) \quad (19)$$

$$= \sum_t [\mathbb{E}_{p_\theta(z_{1:t}|x_{1:t}, a_{1:t})}(\ln q_{\zeta_1}(x_t|x_{1:t-1}, z_{1:t}) + \ln q_{\zeta_2}(a_t|a_{1:t-1}, z_{1:t}))$$

$$- D_{KL}(p_\theta(z_t|x_{1:t}, a_{1:t})||p(z_t))] \quad (20)$$

## 2 Proof for Causal Video Generation with Learned Action Prior

### 2.1 variational Lower Bound

Here we are trying to maximize the joint likelihood of $(x_{1:t}, a_{1:T})$ which is equivalent to maximizing $\ln q_\zeta(x_{1:T}, a_{1:T})$ or $\ln q_\zeta(x, a)$ for better readability. Let's assume $z = [z_1, \cdots, z_T]$ denotes the image latent $z$ variable across all the time-steps and $u = [u_1, \cdots, u_T]$ denotes the action latent $u$ variable across all the time-steps. Both $z_t$s and $u_t$s are independent of across time. From the Causal relationship between $a_t$ and $x_t$ we get:

$$\ln q_\zeta(x_{1:T}, a_{1:T}) \equiv \ln q_\zeta(x, a) = \ln q_{\zeta_2}(a|x)q_{\zeta_1}(x) \quad (21)$$

$$= \ln q_{\zeta_2}(a|x) + \ln q_{\zeta_1}(x) \quad (22)$$

from equation 22, we can derive the lower bound for $\ln q_{\zeta_1}(x)$ as:

$$\ln q_{\zeta_1}(x) = \ln \int_z q_{\zeta_1}(x|z)p(z) \quad (23)$$

$$= \ln \int_z q_{\zeta_1}(x|z)p(z)\frac{p_\theta(z|x)}{p_\theta(z|x)} \quad (24)$$

$$= \ln \left( \mathbb{E}_{p_\theta(z|x)}q_{\zeta_1}(x|z)\frac{p(z)}{p_\theta(z|x)} \right) \quad (25)$$

$$\geq \mathbb{E}_{p_\theta(z|x)} \left( \ln q_{\zeta_1}(x|z)\frac{p(z)}{p_\theta(z|x)} \right) \quad (26)$$

$$= \mathbb{E}_{p_\theta(z|x)}\ln q_{\zeta_1}(x|z) - \mathbb{E}_{p_\theta(z|x)} \left( \ln\frac{p_\theta(z|x)}{p(z)} \right) \quad (27)$$

$$= \mathbb{E}_{p_\theta(z|x)}\ln q_{\zeta_1}(x|z) - D_{KL}(p_\theta(z|x)||p(z)) \quad (28)$$

30 Similar to VG-LeaP, we use RNN architectures $\hat{RNN}_{\zeta_1}$ in Causal-LeaP to recursively predict image

31 frames at each time-step $t$. $\hat{RNN}_{\zeta_1}$ takes the encoded past image $x_{t-1}$, action $a_{t-1}$ and $z_t$ as input.

32 With the recursive behaviour of $\hat{RNN}_{\zeta_1}$, we approximate $\ln q_{\zeta_1}(x|z) \approx \ln q_{\zeta_1}(x_{1:T}|z_{1:T}, a_{1:T-1})$ as:

$$\ln q_{\zeta_1}(x|z) \approx \ln q_{\zeta_1}(x_{1:T}|z_{1:T}, a_{1:T-1}) = \ln \prod_t q_{\zeta_1}(x_t|x_{1:t-1}, z_{1:T}, a_{1:t-1}, \cancel{a_{t:T-1}}) \tag{29}$$

$$= \sum_t \ln q_{\zeta_1}(x_t|x_{1:t-1}, z_{1:t}, a_{1:t-1}) \tag{30}$$

33 In the case of the posterior and learned prior of the image prediction networks $\hat{RNN}_\theta$ and $\hat{RNN}_\phi$

34 respectively in Causal-LeaP, we recursively feed the action image $x_t$ to approximate $z_t$ (in case of

35 learned prior we feed $x_{t-1}$ ). Cause $z_t$s are independent across time, $p_\theta(z|x)$ can be expressed as:

$$p_\theta(z_{1:T}|x_{1:T}) \equiv p_\theta(z|x) = \prod_t p_\theta(z_t|x_t) \tag{31}$$

36 We can rewrite $D_{KL}(p_\theta(z|x)||p(z))$ as:

$$D_{KL}(p_\theta(z|x)||p(z)) \equiv D_{KL}(p_\theta(z_{1:T}|x_{1:T})||p(z_{1:T})) = \int_z p_\theta(z|x)\ln\frac{p_\theta(z|x)}{p(z)} \tag{32}$$

$$= \int_{z_1}\cdots\int_{z_T} p_\theta(z_1|x_1)\cdots p_\theta(z_T|x_{1:T})\ln\frac{p_\theta(z_1|x_1)\cdots p_\theta(z_T|x_{1:T})}{p(z_1)\cdots p(z_T)} \tag{33}$$

$$= \int_{z_1}\cdots\int_{z_T} p_\theta(z_1|x_1)\cdots p_\theta(z_T|x_{1:T})\sum_t \ln\frac{p_\theta(z_t|x_{1:t})}{p(z_t)} \tag{34}$$

$$= \sum_t \int_{z_1}\cdots\int_{z_T} p_\theta(z_1|x_1)\cdots p_\theta(z_T|x_{1:T})\ln\frac{p_\theta(z_t|x_{1:t})}{p(z_t)} \tag{35}$$

37 Since $\int_z p_\theta(z) = 1$ we can further simplify equation 35 as

$$D_{KL}(p_\theta(z|x)||p(z)) = \sum_t \int_{z_t} p_\theta(z_t|x_{1:t})\ln\frac{p_\theta(z_t|x_{1:t})}{p(z_t)} \tag{36}$$

$$= \sum_t D_{KL}(p_\theta(z_t|x_{1:t})||p(z_t)) \tag{37}$$

38 The lower bound of $\ln q_{\zeta_2}(a|x)$ is derived as follows:

$$\ln q_{\zeta_2}(a|x) = \ln \int_u q_{\zeta_2}(a|u,x)p(u|x) \tag{38}$$

$$= \ln \int_u q_{\zeta_2}(a|u,x)p(u|x)\frac{p_\psi(u|a,z)}{p_\psi(u|a,z)} \tag{39}$$

$$= \ln \left( \mathbb{E}_{p_\psi(u|a,z)} q_{\zeta_2}(a|u,x)\frac{p(u|x)}{p_\psi(u|a,z)} \right) \tag{40}$$

$$\geq \mathbb{E}_{p_\psi(u|a,z)} \left( \ln q_{\zeta_2}(a|u,x)\frac{p(u|x)}{p_\psi(u|a,z)} \right) \tag{41}$$

$$= \mathbb{E}_{p_\psi(u|a,z)}\ln q_{\zeta_2}(a|u,x) - \mathbb{E}_{p_\psi(u|a,z)} \left( \ln\frac{p_\psi(u|a,z)}{p(u|x)} \right) \tag{42}$$

$$= \mathbb{E}_{p_\psi(u|a,z)}\ln q_{\zeta_2}(a|u,x) - D_{KL}(p_\psi(u|a,z)||p(u|x)) \tag{43}$$

39 Now combining equation 22, equation 28 and equation 43 we get:

$$\ln q_\zeta(x,a) \geq \mathbb{E}_{p_\theta(z|x)}\ln q_{\zeta_1}(x|z) + \mathbb{E}_{p_\psi(u|a,z)}\ln q_{\zeta_2}(a|u,x) - D_{KL}(p_\theta(z|x)||p(z))$$
$$- D_{KL}(p_\psi(u|a,z)||p(u|x)) \tag{44}$$

40 In the case of action predictor in Causal-LeaP, to predict $a_t$ we use RNN architecture $\hat{RNN}_{\zeta_2}$

41 which takes the past action $a_{t-1}$ and $u_t$ as inputs at time $t$. Thus recursively it builds dependence

upon all past actions $a_{1:t-1}$ and action latent variable $u_{1:t}$. Please note in the case of the action predictor network we do not feed the last image $x_t$ as input. We found that even without the image $x_t$ as input, the action predictor generates accurate approximation of future actions. Thus $\ln q_{\zeta_2}(a|u,x) \approx \ln q_{\zeta_2}(a|u)$ in can be expressed as:

$$\ln q_{\zeta_2}(a_{1:T}|u_{1:T}) \equiv \ln q_{\zeta_2}(a|u) = \ln \prod_t q_{\zeta_2}(a_t|a_{1:t-1}, u_{1:T}) \tag{45}$$

$$= \sum_t \ln q_{\zeta_2}(a_t|a_{1:t-1}, u_{1:t}) \tag{46}$$

In the case of the posterior and learned prior of the action prediction networks $\widehat{RNN}_\psi$ and $\widehat{RNN}_\varphi$ respectively in Causal-LeAP, we recursively feed the action $a_t$ and the image latent variable $z_t$ to approximate $u_t$ (in case of learned prior we feed $(a_{t-1}, z_{t-1})$ ). Cause $u_t$s are independent across time, $p_\theta(u|a,z)$ can be expressed as:

$$p_\psi(u_{1:T}|a_{1:T}, z_{1:T}) \equiv p_\psi(u|a,z) = \prod_t p_\psi(u_t|a_t, z_t) \tag{47}$$

We can rewrite $D_{KL}(p_\psi(u|a,z)||p(u|x))$ as:

$$D_{KL}(p_\psi(u|a,z)||p(z)) \equiv D_{KL}(p_\psi(u_{1:T}|a_{1:T}, z_{1:T})||p(u_{1:T}|x_{1:T})) = \int_u p_\psi(u|a,z) \ln \frac{p_\psi(u|a,z)}{p(u|x)} \tag{48}$$

$$= \int_{u_1} \cdots \int_{u_T} p_\psi(u_1|a_1, z_1) \cdots p_\psi(u_T|a_{1:T}, z_{1:T}) \ln \frac{p_\psi(u_1|a_1, z_1) \cdots p_\psi(u_T|a_{1:T}, z_{1:T})}{p(u_1|x_1) \cdots p(u_T|x_T)} \tag{49}$$

$$= \int_{u_1} \cdots \int_{u_T} p_\psi(u_1|a_1, z_1) \cdots p_\psi(u_T|a_{1:T}, z_{1:T}) \sum_t \ln \frac{p_\psi(u_t|a_{1:t}, z_{1:t})}{p(u_t|x_t)} \tag{50}$$

$$= \sum_t \int_{u_1} \cdots \int_{u_T} p_\psi(u_1|a_1, z_1) \cdots p_\psi(u_T|a_{1:T}, z_{1:T}) \ln \frac{p_\psi(u_t|a_{1:t}, z_{1:t})}{p(u_t|x_t)} \tag{51}$$

Since $\int_u p_\psi(u) = 1$ we can further simplify equation 51 as

$$D_{KL}(p_\psi(u|a,z)||p(u|x)) = \sum_t \int_{u_t} p_\psi(u_t|a_{1:t}, z_{1:t}) \ln \frac{p_\psi(u_t|a_{1:t}, z_{1:t})}{p(u_t|x_t)} \tag{52}$$

$$= \sum_t D_{KL}(p_\psi(u_t|a_{1:t}, z_{1:t})||p(u_t|x_t)) \tag{53}$$

Now combining equation 30, equation 46, equation 37 and equation 53 with equation 44 we get the final expression for the variational lower bound as:

$$\ln q_\zeta(x,a) \geq \mathbb{E}_{p_\theta(z|x)} \ln q_{\zeta_1}(x|z) +$$
$$+ \mathbb{E}_{p_\psi(u|a,z)} \ln q_{\zeta_2}(a|u) - D_{KL}(p_\theta(z|x)||p(z)) - D_{KL}(p_\psi(u|a,z)||p(u|x)) \tag{54}$$

$$= \sum_t [\mathbb{E}_{p_\theta(z_{1:t}|x_{1:t})} \ln q_{\zeta_1}(x_t|x_{1:t-1}, z_{1:t}, a_{1:t-1}) + \mathbb{E}_{p_\psi(u_{1:t}|a_{1:t}, z_{1:t})} \ln q_{\zeta_2}(a_t|a_{1:t-1}, u_{1:t})$$

$$- D_{KL}(p_\theta(z_t|x_{1:t})||p(z_t)) - D_{KL}(p_\psi(u_t|a_{1:t}, z_{1:t})||p(u_t|x_t))] \tag{55}$$

## 3    Robot Autonomous Motion Dataset: RoAM

The Robot Autonomous Motion (RoAM) dataset is a unique, publicly available resource offering synchronized and time-stamped camera motion data along with video data. It comprises 50 long video sequences collected over 7 days in 14 different indoor spaces, capturing various human activities from the ego-motion perspective of a mobile robot. RoAM provides stereo image pairs, depth

maps, LiDAR scans, IMU data, odometry, and timestamped control actions. The dataset has been processed into accessible formats, with images in .png format and depth maps in 32-bit single-channel .tiff format. Control action data, including normalized forward velocity and rotation rate, are saved in corresponding .txt files. We followed the instructions provided in the GitHub repository (https://github.com/meenakshisarkar/RoAM-dataset.git). to prepare the dataset as a TensorFlow dataset object. This involved sampling 3,07,200 random video-action sequences of length 25 from 45 training videos, with the remaining 5 videos serving as the test set. Images were cropped and downsampled to 64x64x4, preserving the aspect ratio. The repository also includes scripts for reading TFRecord files.

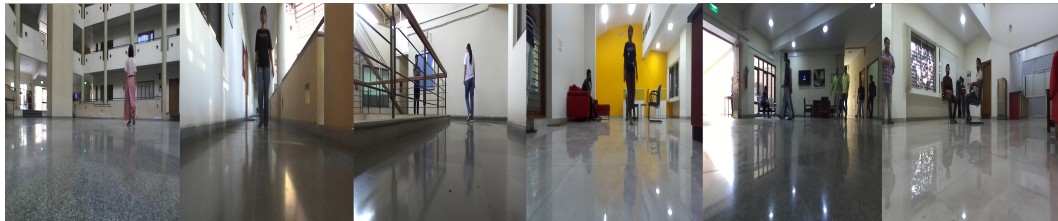

Figure 1: Sample image frames from the RoAM dataset depicting different indoor spaces and backgrounds with various human actions and motions.

## 4 Network Architecture and Training details:

All the models are conditioned on past 5 frames to predict the next 10 frames during training. At test time, we increase the no of predicted frames to 20.

### 4.1 Model: SVG-lp

We used the same architecture for SVG-lp as given in Denton and Fergus (2018). We used the VGG16 architecture for training our image encoder. The size of the latent variable $z$ is taken as 128. The hidden dimension for the vgg16 image encoder is taken as 256 and we used 512 lstm cells in the recurrent block of our posterior and prior modules. Similar to Denton and Fergus (2018), we used 2 layers of recurrent cells, each having 512 LSTM cells in our image prediction network.

We trained the SVG-lp on RoAM using the ADAM Kingma and Ba (2015) optimizer with the learning rate of 0.0001. $\beta$ for the KL divergence term was set at 0.001, with $\beta 1 = 0.9$ for the ADAM optimizer. The batch size was kept at 64. The model was trained on two Tesla v100 GPUs using the distributed training API of tensorflow.

### 4.2 Model:VG-LeAP

VG-LeAP uses the same architecture and training parameters as SVG-lp as discussed in the previous subsection. It also uses the VGG16 architecture for training our image encoder to encode images as a vector of size 256. The size of the latent variable $z$ is also taken as 128 and we used 512 lstm cells in the recurrent block of our posterior and prior modules. Similar to SVG-lp, we use 2 layers of recurrent cells, each having 512 LSTM cells in our image prediction network. The only extra hyper-parameters in VG-LeAP comes from the action encoder and predictor network. The action encoder encodes the 2 dimensional input action to an encoded dimension of 16. In the case of action encoding, we upsample the action to a higher dimensional manifold for better interaction between the encoded image and action data in the approximation of the latent variable $z_t$. We use 2 layers of LSTM cells, each having 32 LSTM cells in the action prediction network.

We trained the VG-LeAP on RoAM using the ADAM Kingma and Ba (2015) optimizer with the learning rate of 0.0001. $\beta$ for the KL divergence term is set at 0.001, $\beta_a = 0.0001$, and $\beta 1 = 0.9$ for

94 the ADAM optimizer. The batch size was kept at 64. The model is trained on two Tesla v100 GPUs
95 using the distributed training API of Tensorflow.

### 4.3 Model:Causal-LeAP

97 Causal-LeAP uses the same vgg16 image encoding architecture as VG-LeAP and SVG-lp. The size
98 of the latent variable $z$ is also taken as 128 and the hyper-parameters for the image prior, posterior and
99 prediction network are kept same as VG-LeAP and SVG-lp for better comparison of performances.
100 The hyper-parameters action encoder and predictor networks in Causal-LeAP is same as VG-LeAP.
101 In Causal-LeAP we have the additional Action Posterior and Prior modules which uses a single layer
102 of 32 LSTM cells to establish the recurrent relationship between the past action $a_t$ and latent image
103 variable $z_t$ values in approximation of the latent action variable $u_t$

104 We trained the Causal-LeAP with the same training parameter as VG-LeAP on RoAM using the
105 ADAM Kingma and Ba (2015) optimizer with a learning rate of 0.0001 and batch size of 64 except
106 in Causal-LeAP the $\beta$ associated with the image KL divergence is 0.0001. $\gamma = 0.0001$ for the KLD
107 associated with the action latent variable. The model is also trained on two Tesla v100 GPUs using
108 the distributed training API of Tensorflow.

### 4.4 Model: RIVER and RAFI

110 Our overall architecture for both river and rafi remained same as given in Davtyan, Sameni, and
111 Favaro (2023). We first trained a VQGAN Esser, Rombach, and Ommer (2021) using using the
112 taming-transformers library, for both RIVER and RAFI we used the same pretrained VQGAN. For
113 training of VQGAN we first sampled 2 videos from each video sequence and trained the VQGAN
114 architecture for 8 iterations. Then we further trained it for 2 more iterations but with 10 frames being
115 sampled from each Video Sequence. In our training set of VQGAN we included images both from
116 our test set and our train set.

117 The only difference in architecture between RAFI and RIVER is in the dimensions of the input to the
118 vector field regressor model, for river its input state size was kept as 8 and input state resolution at 4.
119 For RAFI these parameters were kept as 10 (due to 2 additional channels generated by concatenating
120 action maps) and 4 respectively. In order to train the vector field regressor model we used a batch size
121 of 16. Both models were trained for 450k iterations with AdamW optimizer, with initial lr as $10^{-4}$
122 and weight decay as $5.10^{-6}$. Similar to Davtyan, Sameni, and Favaro (2023) we used A learning
123 rate linear warm up with $7.5K$ iterations followed by a square root decay schedule. For both of these
124 models, we kept the no. of conditioning frames to 5 and total no. of frames to generate to 10.

### 4.5 Model: SRVP

126 We trained SRVP following the training parameters given in Franceschi et al. (2020) for training
127 Human3.6M dataset as that was the only dataset that closely resembled RoAM. We trained SRVP on
128 RoAM dataset using the distributed training API of PyTorch Paszke et al. (2019) on two NVIDIA
129 3090Ti GPUs with a batch size of 32. In a distributed training setup, SRVP took 36 hours to train on
130 RoAM dataset for 525000 iterations. We found that increasing the batch size to 64 for SRVP resulted
131 in exponentially increasing the training time for the model from 1.5 days to several days(4.5). Thus
132 we kept 32 as our batch-size for SRVP to accommodate our limitation in computational resources.

133 The learning rate is kept at 0.0001 and the number of Euler steps is kept as 2. The rest of the training
134 parameters are kept as same as training the Human3.6M in Franceschi et al. (2020).

### 4.6 Model: ACPNet

136 We also trained ACPNet on the RoAM dataset following the same training hyper-parameters as
137 Sarkar et al. (2023).

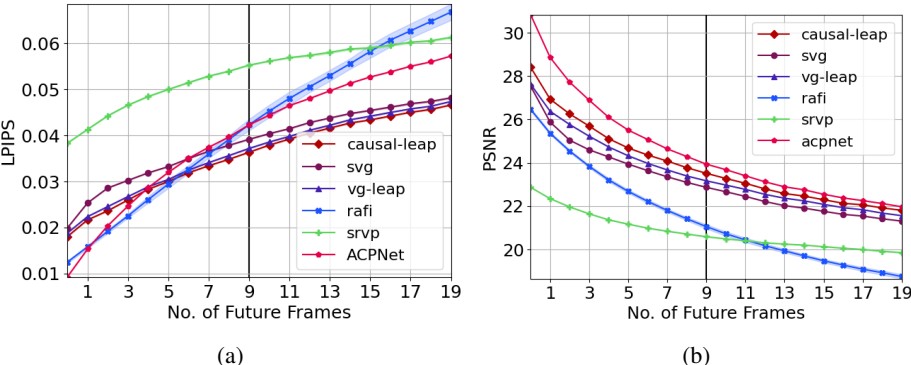

(a)  (b)

Figure 2: Fig. 2a (lower is better), and 2b(higher is better) showing the average quantitative performance of Causal-LeAP, VG-LeAP, SVG (SVG-lp), RAFI, SRVP, and ACPNet for 20 different sampling on predicting 20 future image frames from past 5 conditioning frames. In all the quantitative performance metrics, Causal-LeAP model outperforms the other 5. In case of LPIPS values for RAFI and ACPNet, we can see that both these models start much better than Causal-Leap, however as time passes, both start performing much worse than LeAP models.

## 5   Extended Results and Additional Generated Images

We have provided the LPIPS, and PSNR plots for all 6 models: Causal-LeAP, VG-LeAP, SVG-lp, RAFI, SRVP Franceschi et al. (2020) and ACPNet in Fig 2a and 2b respectively. It can be observed from the plots that SRVP does not perform well on RoAM dataset. From the extended FVD score table in 1 we can see that the FVD score for SRVP is 596, which is relatively higher compared to SVG and other LeAP models. We attribute this to the poor training of SRVP model on RoAM data. There might be some combination of the hyper parameter tuning that might lead to better performance on RoAM, however, the large computational requirements of SRVP limited our scope for such explorations. This also raises an important question of how efficient frameworks like SRVP Franceschi et al. (2020) on modelling partially observable video data such as RoAM and warrent further investigation.

We have also added raw-generated image frames by Causal-LeAP, VG-LeAP, SVG-lp, RAFI, and RIVER at Fig 3, 4,5, 6,7, 8. Additional generated videos can be found at the following link: https://sites.google.com/view/learned-action-prior/home.

Table 1: FVD Score

| Model | Score |
|---|---|
| Causal-Leap | $514.65 \pm 3.37$ |
| Svg-Leap | $539.29 \pm 1.94$ |
| Vg-Leap | $481.15 \pm 2.39$ |
| **RIVER (BEST)** | **$284.46 \pm 3.21$** |
| **RAFI** | **$288.23 \pm 4.39$** |
| SRVP | $596.68 \pm 2.82$ |
| ACPNET | 908.36 |

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

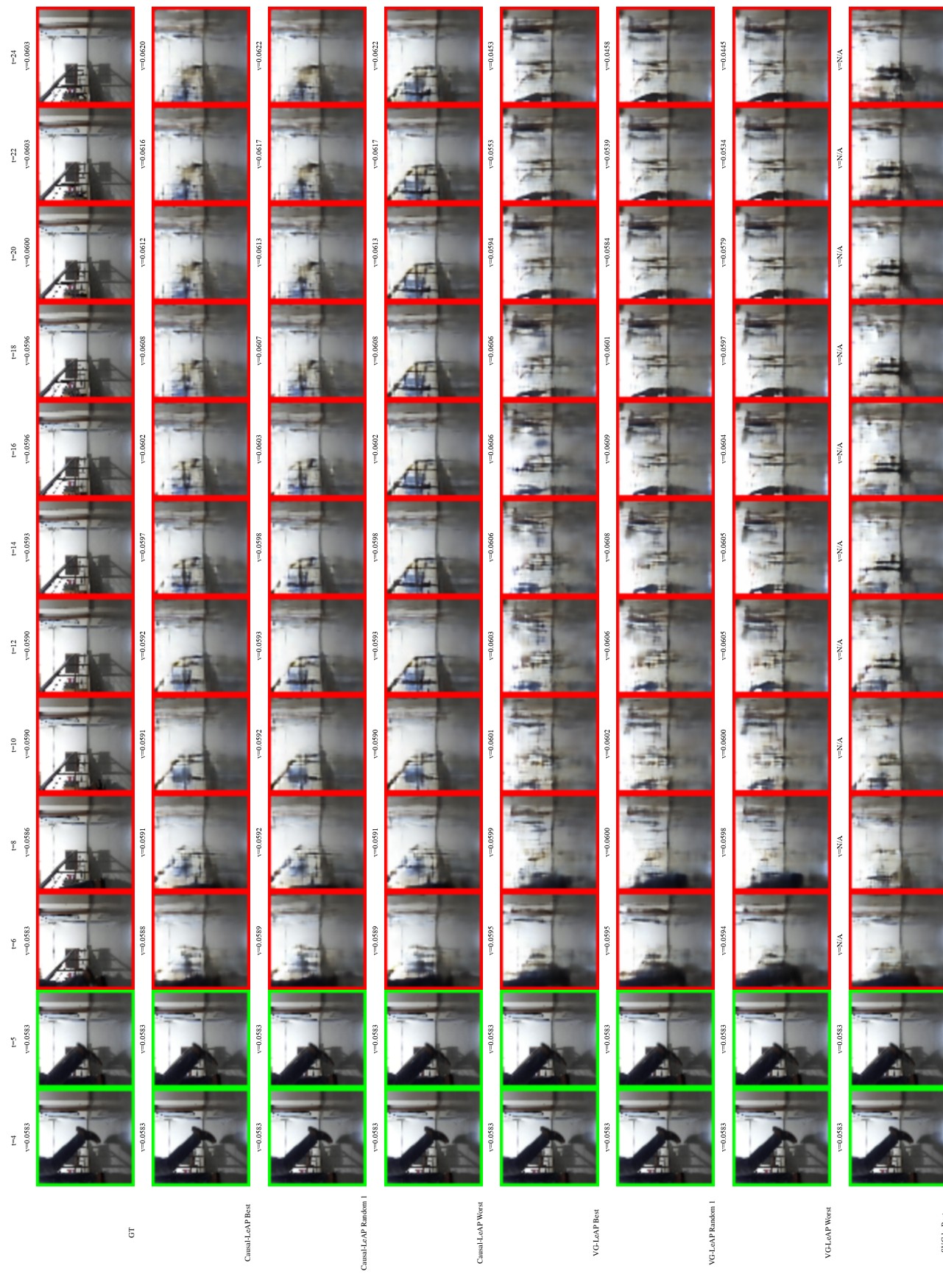

Figure 3: Samples from Causal-LeAP, VG-LeAP, and SVG-lp compared to Ground Truth. For each framework, we show the best, worst, and one random samples based on VGG cosine similarity. Each framework predicts 20 future image-action pairs conditioned on 5 past image-action observations. The normalised forward velocity in case of Ground Truth, Causal-LeAP and VG-LeAP are denoted at the top of the frames.

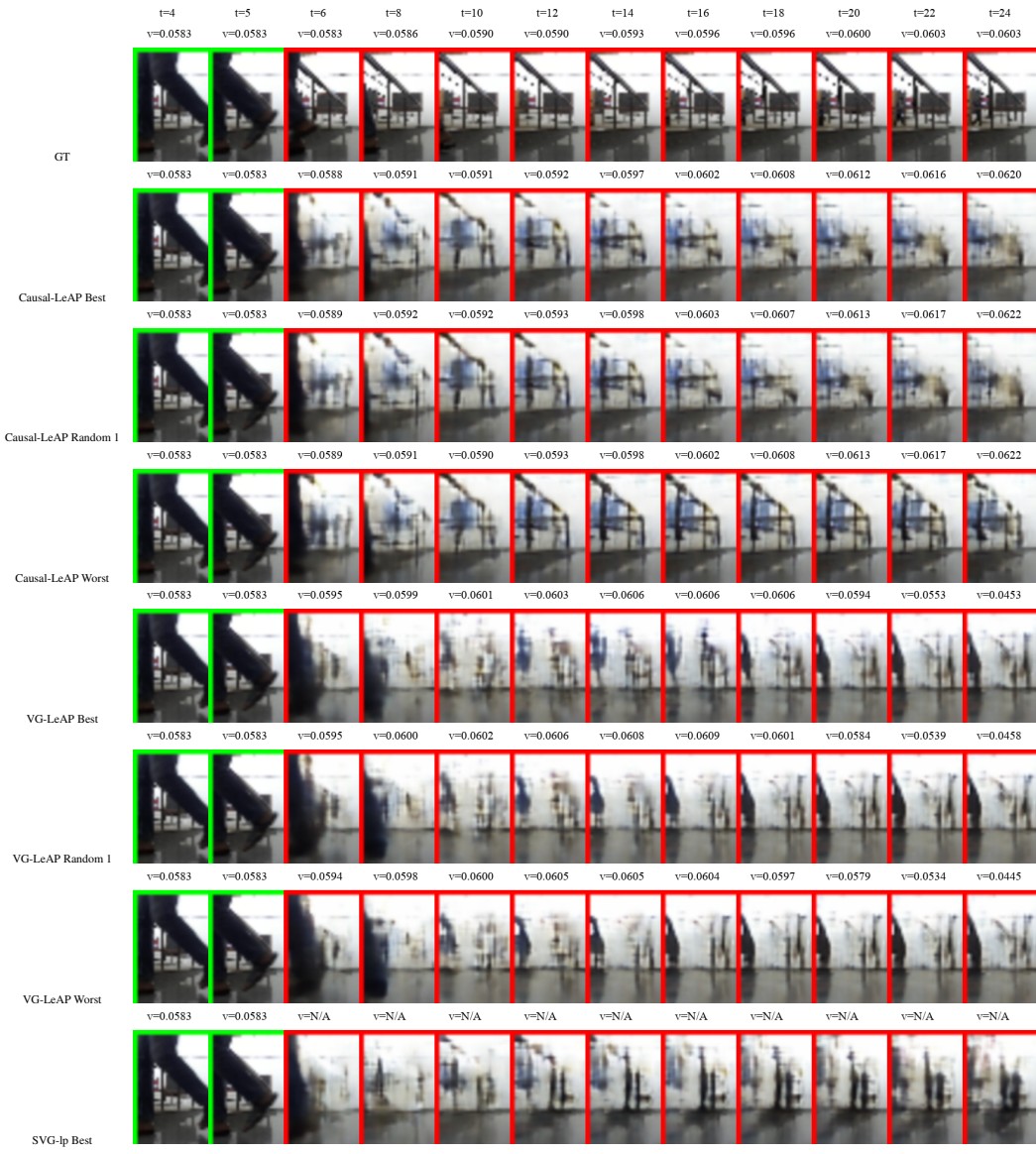

Figure 4: Zoomed samples from Causal-LeAP, VG-LeAP, and SVG-lp compared to Ground Truth. For each framework, we show the best, worst, and one random samples based on VGG cosine similarity. Images are zoomed using bilinear extrapolation for better visibility.

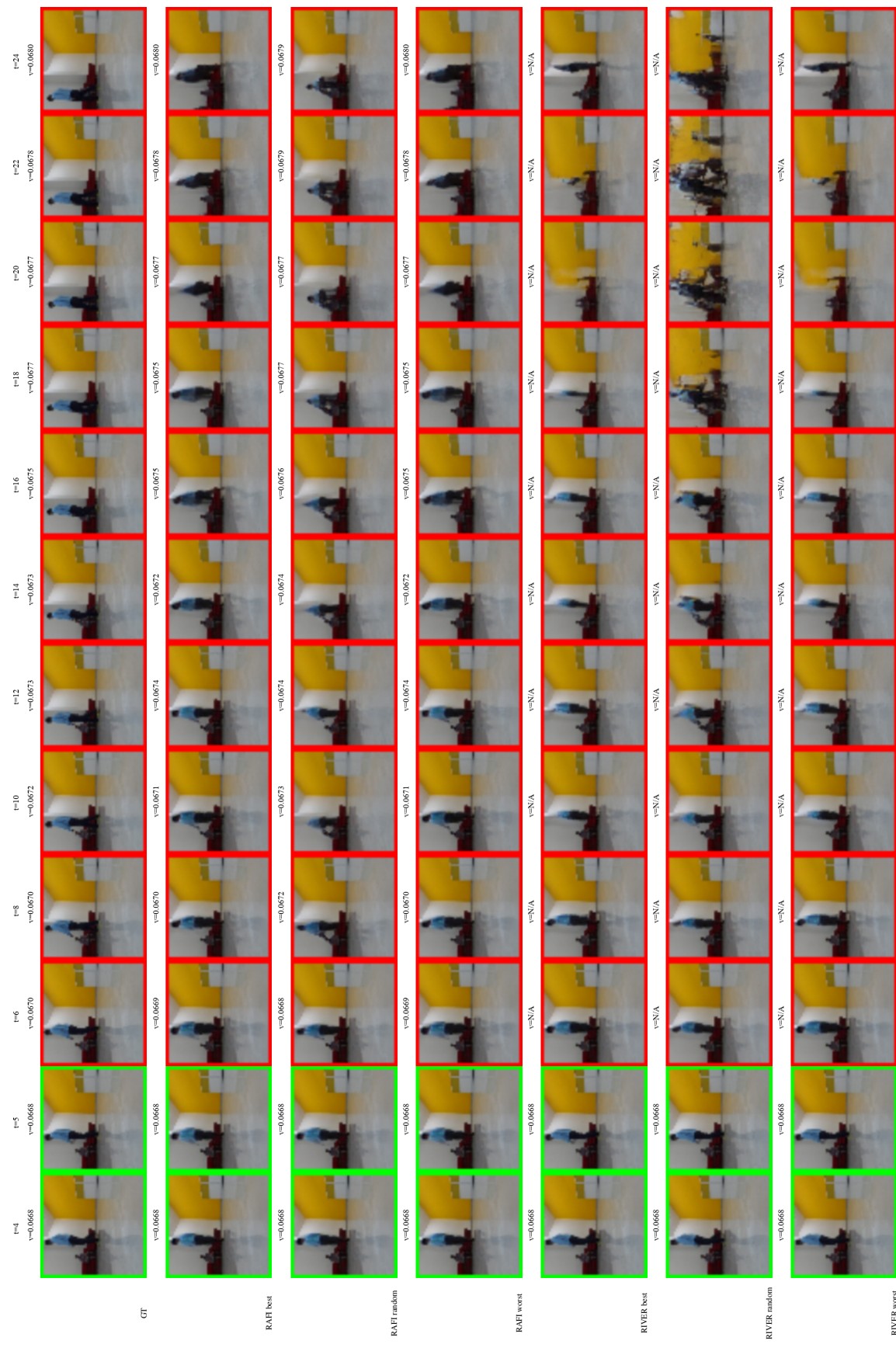

Figure 5: selected samples from RAFI and RIVER along with GT. For each framework, we show the best, worst, and one random samples based on VGG cosine similarity. Each framework predicts 20 future image-action pairs conditioned on 5 past image-action observations. Here also the normalised forward velocities are denoted at the top of the frames similar to Fig. 3.

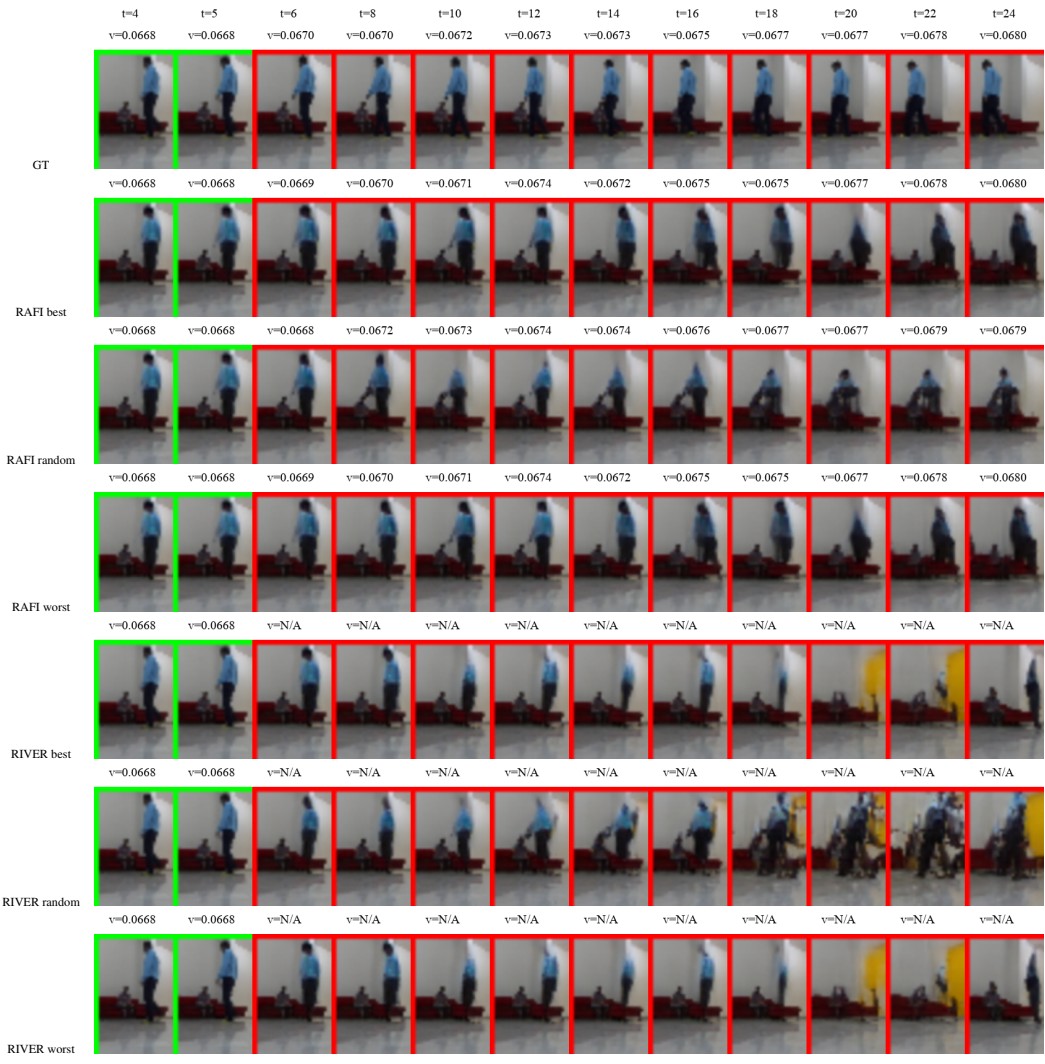

Figure 6: Zoomed samples from RAFI and RIVER along with GT. For each framework, we show the best, worst, and one random samples based on VGG cosine similarity. Images are zoomed using bilinear extrapolation for better visibility.

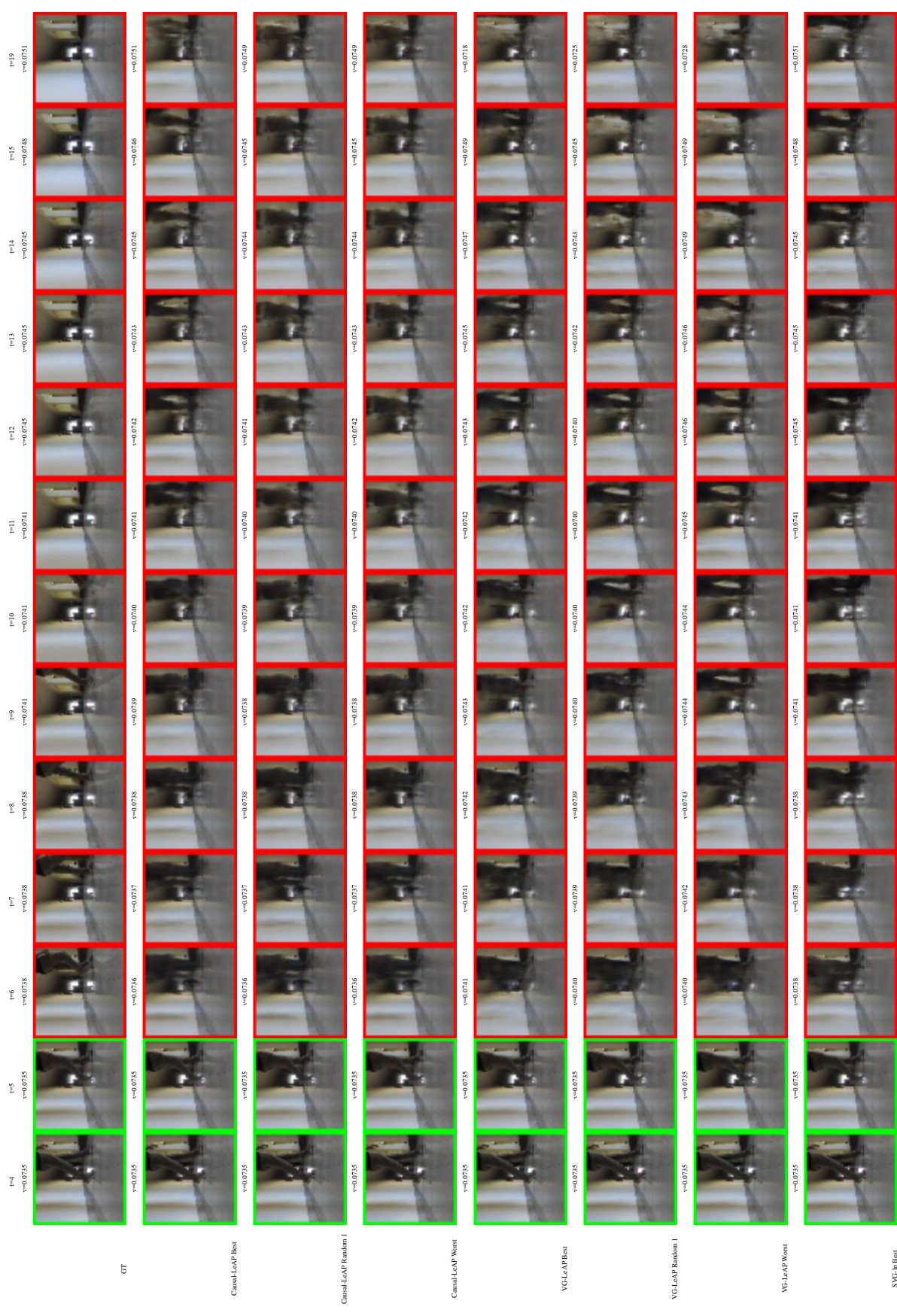

Figure 7: Ablation study samples from Causal-LeAP, VG-LeAP, and SVG-lp compared to Ground Truth. For each framework, we show the best, worst, and one random samples based on VGG cosine similarity. Each framework predicts 15 frames into the future from past 5 frames at 0.5 fps$_{\text{train}}$ or $\Delta t_{\text{test}} = 2 \times \Delta t_{\text{train}}$. The normalised forward velocities are denoted at the top of the frames.

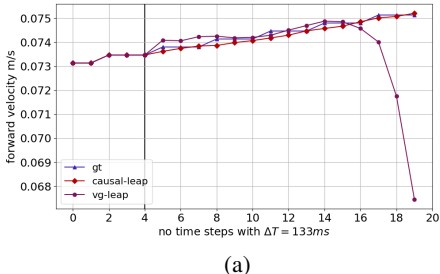
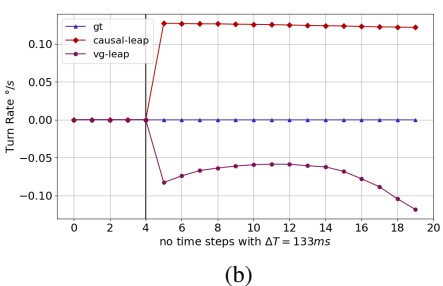

(a)                                                    (b)

Figure 8: Fig. 8a and 8b shows the predicted forward velocity and turn rates from Causla-LeAP and VG-LeAP along with GT for the ablation study video frame presented in Fig. 7. Here also we can see that predictions from VG-LeAP starts to diverge after timestep $t = 14$.