# OpenReview forum: "Video Generation with Learned Action Prior"
_ICLR.cc/2025/Conference — Submitted to ICLR 2025_

### Official Review · Reviewer_MP4m · 2024-10-28

**Soundness:** 3
**Presentation:** 2
**Contribution:** 2
**Rating:** 3
**Confidence:** 3

**Summary:**

This paper tackles the partially observable video prediction problem, where the camera is in motion, by incorporating both camera movement and action. The authors propose three models: (1) VG-LeAP, which treats the image-action pair as a state from a single latent stochastic process; (2) Causal-LeAP, which learns a separate action prior conditioned on the observed images and action history; and (3) RAFI, which integrates the augmented image-action state with a conditional flow matching framework. Empirical results on the RoAM dataset demonstrate the effectiveness of these models in addressing partially observable video generation.

**Strengths:**

1. The paper addresses an important and interesting problem in video generation—partially observable video prediction.
2. The paper introduces three models based on variational frameworks and conditional flow matching.

**Weaknesses:**

1. The paper lacks a comparison with prior works that model camera motion in video generation. Several existing studies incorporate camera motion information [1-6]. The authors should discuss how their methods differ from these works and clarify the advantages of their approach.
2. Experiments are limited to the RoAM dataset, which includes camera action annotations. Testing the models on additional datasets and demonstrating generalization to video data without explicit camera action annotations (like A2D2 [7]) would strengthen the effectiveness of proposed models.
3. The rationale for proposing three distinct models is unclear. The authors should explain the specific advantages, disadvantages and computational complexities of each model and provide guidance on when to use each one.
4. The frames shown in Figures 6 and 8 are too blurry, hindering a proper assessment of video prediction quality. Providing zoomed-in key areas or video samples would improve evaluation clarity.

[1] Guo, Yuwei, et al. "AnimateDiff: Animate Your Personalized Text-to-Image Diffusion Models without Specific Tuning." The Twelfth International Conference on Learning Representations.

[2] Wang, Xiang, et al. "Videocomposer: Compositional video synthesis with motion controllability." Advances in Neural Information Processing Systems 36 (2024).

[3] Wang, Zhouxia, et al. "Motionctrl: A unified and flexible motion controller for video generation." ACM SIGGRAPH 2024 Conference Papers. 2024.

[4] He, Hao, et al. "Cameractrl: Enabling camera control for text-to-video generation." arXiv preprint arXiv:2404.02101 (2024).

[5] Yang, Shiyuan, et al. "Direct-a-video: Customized video generation with user-directed camera movement and object motion." ACM SIGGRAPH 2024 Conference Papers. 2024.

[6] Xu, Dejia, et al. "CamCo: Camera-Controllable 3D-Consistent Image-to-Video Generation." arXiv preprint arXiv:2406.02509 (2024).

[7] Geyer, Jakob, et al. "A2d2: Audi autonomous driving dataset." arXiv preprint arXiv:2004.06320 (2020).

**Questions:**

1. What are the differences between the proposed methods and prior works listed in Weakness? What are the advantages of the proposed method?
2. What are the connections between the three proposed models? What is the purpose of proposing three models? What are the advantages and disadvantages of each model?
3. Can the proposed method generalize to video data without explicit camera action annotations?

---

> ### Author Response · Authors · 2024-11-30
> **Response 1/2**
>
> Thank you for your thoughtful questions and for reviewing our work. Below are our detailed responses:
> ### **1. Comparison to previous works:**
> We have modified the Introduction section in our revised paper to account for the aforementioned clarification while also conforming to the given page limit. The paragraph from Line no 64-73 specifically compares our current work with latent flow-based and diffusion-based video generation models.
>
> Video diffusion models with camera control like [1] AnimateDiff, [2] Videocomposer, [3] Motionctrl, and [4] Direct-a-video leverage textual instructions and occasionally camera parameters like pan and zoom (in Direct-a-video) to generate high-fidelity videos. However, all these models assume prior knowledge of desirable camera movement. Similarly, [5] CamCo and [6] CameraCtrl utilize Plücker embeddings to condition generated videos on camera motion, also assuming predetermined camera movement.
> In contrast, our work models camera dynamics from observed image states and predicts future camera movement and images in unpredictable environments like busy roads or crowded spaces. Our approach has significant potential applications in mobile robotics, particularly in safe navigation and obstacle avoidance.
> ### **2 distinct models:**
>  We believe our modified introduction section from line 74-93 in the revised paper clarifies the need and distinction between the 3 models VG-LeAP, Causal-LeAP and RAFI. Out of the three models, VG-LeAP and Causal-LeAP are variational inference models. VG-LeAP assumes conditional independence, positing that image and action are generated from the same composite latent stochastic process, while Causal-LeAP models the causal relationship between camera motion and observed image frames using two distinct learned priors.
>
> Both models are suitable for autonomous motion planning in robotics and autonomous driving, presenting no significant computational overhead compared to the seminal SVG-LP framework. However, Causal-LeAP's ability to build a causal relationship between camera/robot action and learn a prior based on observed/predicted image data renders it particularly valuable to the RL and robotics community, as demonstrated in our control action plots in Figures 6 and 7b, where Causal-LeAP predicts the robot's avoidance maneuver.
>
> To our knowledge, Causal-LeAP is the first model to model the causal relationship between camera motion and observed image frames in video prediction tasks. We believe this theoretical framework could potentially be extended to other data modalities like text in future research.
>
> RAFI is based on the conditional independence condition and built upon the Conditional Flow Matching framework. We introduced RAFI to demonstrate how our theoretical framework for variational models can be extended to more recent methodological concepts like Flow matching. However, RAFI is computationally significantly more complex than VG-LeAP and Causal-LeAP, with limited practical applicability for real-time predictions in safety-critical domains such as robotics or autonomous driving. Despite these limitations, we demonstrate that latent flow-based models can simultaneously learn camera dynamics and video spatio-temporal dynamics, potentially motivating future research into causal frameworks.
> ### **3 generalize to video data without explicit camera action annotations**
> Yes all our models will work on video without explicit camera annotation, we just have to set the weights of the associated component of the loss function to the Action Prediction module to be 0. However the primary objective of our work was to learn and predict the unknown camera motion dynamics from the observed images and thus not having camera annotation will defeat the purpose.
> ### **4 Blurry Images in 6 and 8:**
> As all our frameworks are trained and evaluated on images of resolution $64\times 64$, we could not generate very sharp or high-resolution images. This limitation stems from our computational resource constraints. Video prediction frameworks require significantly higher computational resources to train and evaluate, and we primarily used NVIDIA 3090 systems for our research.
> With sufficient computational resources, all our frameworks can be trained on high-resolution images to generate more sharper predictions.
>
> We hope this clarifies the key distinctions and contributions of our work. Thank you again for your thoughtful review and questions and we remain open to your feedback.

---

> > ### Author Response · Authors · 2024-11-30
> > **Response 2/2**
> >
> > #### [1] Guo, Yuwei, et al. "AnimateDiff: Animate Your Personalized Text-to-Image Diffusion Models without Specific Tuning." ICLR
> > #### [2] Wang, Xiang, et al. "Videocomposer: Compositional video synthesis with motion controllability." NeurIPS 36 (2024).
> > #### [3] Wang, Zhouxia, et al. "Motionctrl: A unified and flexible motion controller for video generation." 2024.
> > #### [4] Yang, Shiyuan, et al. "Direct-a-video: Customized video generation with user-directed camera movement and object motion." 2024.
> > #### [5] Xu, Dejia, et al. "CamCo: Camera-Controllable 3D-Consistent Image-to-Video Generation." (2024).
> > #### [6] He, Hao, et al. "Cameractrl: Enabling camera control for text-to-video generation."

---

> > > ### Author Response · Authors · 2024-12-01
> > > **Gentle Reminder for feedback to our reply**
> > >
> > > Dear reviewer MP4m,
> > >
> > > Thank you for your time in providing reviews for our paper. As the final deadline is approaching, we would like to gently remind you that we have submitted our rebuttal addressing the key concerns and questions you raised regarding our motivation for proposing 3 different models and their respective advantages and disadvantages and applications, the generalisation of our models to other video data without action annotation along with a detailed discussion of prior action conditioned diffusion and flow based models as citated in your feedback.
> > >
> > > We have provided comprehensive explanations to clarify these points along with a revised version of our paper and look forward to your feedback.
> > >
> > > Thank you for your time and consideration.

---

> > > > ### Comment · Reviewer_MP4m · 2024-12-02
> > > >
> > > > After reading the comments from other reviewers and the rebuttal, I think the key weaknesses of this work are as follows:
> > > >
> > > > 1. Dataset limitations: The experimental datasets are limited to RoAM, despite claims that the proposed method advances autonomous navigation and robotic planning and could generalize to video data without camera action annotations. Evaluations on more datasets are necessary to support this claim.
> > > > 2. Insufficient comparisons with video generation methods: While these methods may not explicitly model camera dynamics, they are capable of generating videos with camera dynamics. A comparison under the same experimental conditions would better demonstrate the strengths of the proposed method.
> > > > Given these concerns, I do not think the work in its current form is ready for acceptance and will maintain my original rating.

---

> > > ### Comment · Reviewer_MP4m · 2024-12-02
> > >
> > > Thank you for your response. While the explanation addressed some of my concerns, I still have a few questions:
> > >
> > > 1. I understand that the proposed method explicitly models camera dynamics from observed image states to predict future camera movements and images, whereas previous methods rely on predetermined camera movements for control. However, some video generation models (e.g., AnimateDiff) can generate videos with camera motion because they are trained on large-scale datasets that inherently include camera motion. In such models, camera dynamics and content changes are implicitly learned in the latent features, and camera dynamics can be decoded from these features. What are the advantages of the proposed method compared to these implicit learning models? Additionally, how are these advantages validated in the experiments?

---

> ### Author Response · Authors · 2024-12-02
> **Response to Reviewer's Reply:**
>
> Thank you for your follow-up comments and for carefully considering our rebuttal. We appreciate the opportunity to address the remaining concerns and further clarify our contributions.
>
> **Advantages of Explicit Modeling of Camera Dynamics**
> AnimateDiff and other state-of-the-art methods implicitly learn camera actions due to their training on large-scale datasets containing diverse camera movements. While these approaches are effective, our framework provides several key advantages:
>
> * We propose two theoretical frameworks to generalize camera motion conditioning and prediction in video prediction frameworks. Our framework is general and can be seamlessly extended to models like AnimateDiff. Incorporating explicit camera movements into such models using our proposed theoretical framework is expected to enhance their performance. This assertion is supported by our experiments, where we show that explicitly modeling camera motions yields superior results compared to implicit methods.
>
> * To demonstrate the benefits of explicit modeling, we used the SVG-LP model as a baseline. This model implicitly incorporates actions as an extended state, following the approach of [cite SVG paper]. Our experiments reveal that both models developed using our explicit modeling framework—**Causal-LeAP** and **VG-LeAP**—significantly outperform SVG-LP. This empirical evidence reinforces the conclusion that explicit modeling is more effective than implicit methods.
>
> * In robotics, it is very costly to create a large-scale dataset. For example, even the Kitti dataset only has 50 training video sequences. Owing to this limitation, it is crucial to explicitly predict and learn a prior over actions and camera movements.
>
> **Dataset Limitations**
> The lack of diverse datasets is a significant limitation, as **RoAM** is currently the only publicly available dataset offering synchronized image-action pair data for training and validation, restricting our experimental scope. A broader collection of datasets, such as video data from cameras mounted on dynamic robotic manipulators or autonomous vehicles, would enable more comprehensive analysis and generalization.
>
> **Addressing Concerns on Comparisons with Video Generation Methods**
> We propose two theoretical frameworks for flow matching models and compare them with four other variational models in addition to our three proposed models. By extending our theoretical framework to two flow-based models, SVG-LP and RIVER, we provide a foundational contribution that offers novel insights into explicitly modeling camera motion dynamics. Our work serves as a stepping stone for future researchers to expand and apply this framework across diverse video generation models, including diffusion-based approaches like AnimateDiff, with the goal of advancing understanding and methodologies in video prediction techniques.
>
> We appreciate the reviewer's feedback and believe that this clarification highlights the unique capabilities and contributions of our framework in addressing real-time video prediction challenges.

---

### Official Review · Reviewer_1jrp · 2024-11-02

**Soundness:** 2
**Presentation:** 2
**Contribution:** 2
**Rating:** 3
**Confidence:** 5

**Summary:**

This paper extended the stochastic video generation method with explicit encoding of camera motion dynamics, and then proposes three models , 1) SVG-LP extended with image-action joint states, 2) SVG-LP with disentangled image and action states (with a predefined causal dependency), and 3) RIVER jointly conditioned by the image-action states. This paper compares the proposed models in a action-labeled robotic video dataset named RoAM, and shows certain improvements than existing approaches that do not include camera motion dynamics.

**Strengths:**

- It shows that combining the camera motion dynamics with visual dynamics will help the video prediction as well as the action prediction. This idea is straightforward but is useful in many cases, such as the design of the world models in embodied agents.
- The paper is easy to follow, and the supplementary materials seem comprehensive.

**Weaknesses:**

Novelty
- The main weakness of this paper is lack of novelty. Indeed, applying camera motion dynamics to condition video generation is not a new story. Recent studies have even tried to customize video generation with user-directed camera movement (Direct-a-video, siggraph'24), or abstract textual motion descriptions (LEGO, eccv'24). In this case, multimodal training of actions and images to the basemodel such as SVG-LP may not meet the expectations of the audience in recent research communities. However, the modification of RIVER is quite trivial and the gain is quite marginal. It was expected that a good action interaction scheme with visual space would lead to a powerful world model that can predict faithful future actions along with physically reliable future frames. The proposed method did not show such kind of potential, according to its results.

Dataset Limitations
- Limited diversity: The RoAM dataset is just about several indoor scenes with a specific robot and features mainly corridors, lobby space, staircases with human movements like walking and sitting. This limited setting may not fully represent the wide variety of real-world scenarios with moving cameras. It would be beneficial to test on a more diverse set of datasets that include a broader range of environments and camera movements.
- Data size and complexity: While the dataset contains a significant number of video sequences (more than 300k), each sequence has only 25 frames of image size 64 × 64 × 4 (again, the diversity of scenarios is limited). This relatively small frame size and limited sequence length (just 1 second may not capture large motions) may not capture all the complex spatio-temporal dynamics that occur in real videos. Larger frame sizes and longer sequences could provide more detailed information for the models to learn from, potentially leading to better performance and more accurate video generation.

Comparison with State-of-the-Art
- Poor visual quality and overfitting issues. The reported qualitative results are of poor visual quality, making it hard to convince that the proposed action prior indeed works. Moreover, the action prediction errors are surprisingly small. Are the GT action values small, or the overfitting issue just happened?
- Incomplete benchmarking: The paper compares the proposed models with a few existing methods such as SVG-lp, RIVER, and ACPNet (cannot find its reference though). However, there are many other recent and relevant state-of-the-art video generation models that are not included in the comparison. For example, some of the latest diffusion-based models or other advanced variational frameworks may have different approaches to handling camera motion and image-action dynamics.
- Lack of ablation studies on key components: While an ablation study is conducted to compare the performance of Causal-LEAP, VG-LeAP, and SVG, more ablation studies on other key components of the models could be useful. For example, analyzing the impact of different encoding and decoding strategies, the role of the latent variables in capturing image-action dynamics, or the effect of the conditional flow matching in RAFI on the overall performance would provide a deeper understanding of the models and help in identifying areas for improvement.

Minor Issue
- The citation in the paper may use the wrong latex command, using \citep{} instead of \cite{}.
- The font sizes in the plots are too small, especially Fig. 4 and 5, 7.

**Questions:**

Please check the questions in the section of the weaknesses.

---

> ### Author Response · Authors · 2024-12-01
> **Response 1/2**
>
> We appreciate the reviewer's valuable feedback and thoughtful insights. Their comments highlighting the significance of our research in embodied AI and mobile robotics are particularly encouraging. We would like to further emphasize that motion planning in unknown environments with multiple moving obstacles has been a primary motivation behind our Causal-LeAP and VG-LeAP models. We would like to draw the reviewer’s attention to the following points.
>
> ### **1. Novelty:**
> We respectfully disagree with the reviewer’s assertion regarding the lack of novelty in our work. Our contributions represent a substantive departure from current state-of-the-art techniques in understanding spatio-temporal dynamic in video in the presence of camera motion. We have also modified the Introduction section (line 046-093) in our revised paper for clarification while also conforming to the given page limit.
>
> We introduced two new theoretical frameworks (i) Conditional Independence and (ii) Causal Dependence, that not only incorporate actions into video prediction but simultaneously predict future actions. We have provided detailed discussion regarding these two theoretical frameworks in line 74-92.
>
> In stochastic settings such as on a busy road or crowded spaces we cannot rely on the assumption of the knowledge of the pre-computed desired camera movement (Direct-a-video, Videocomposer). In these scenarios an ideal approach requires the ability to learn and predict platform actions based on past and predicted image frames, and vice versa which is exactly what all our proposed models VG-LeAP, Causal-LeAP and RAFI do.
>
>  Under the assumption of Conditional Independence, we assume that image and action are generated together from a composite latent process. VG-LeAP and RAFI were designed upon this assumption  whereas in case of Causal Dependence in Causal-LeAP, we learn the causal relationship between camera motion and observed image frames using two distinct learned priors.
> We have also provided a more clear comparison to other existing video prediction frameworks in partially observable scenarios to our current work in Lines 46-57.
>
>   - **Novelty of Causal-LeAP:** To our knowledge Causal-LeAP is the first model that tried to model the causal relationship between the camera motion and the observed image frames in the dynamic camera scenario. We have further derived a novel ELBO loss in Eq 13 for our proposed causal model. The causality assumption and separate latent space model for action prediction hold significant implications for end-to-end image-based reinforcement learning motion planners in robotics, potentially allowing direct integration of the learned action prior into RL control agents.
>
>  - **Other camera control models:** : We have added 2 new paragraphs in our introduction (Lines 57-73 of our revised paper) to address the reviewers concern regarding the distinction between our proposed frameworks and other prior works  in action-conditioned video prediction/generation like Direct-a-Video or Videocomposer.
> ### **2 Dataset Limitations:**
>  - **Data Diversity:** Currently RoAM is the only publicly available dataset offering synchronised image-action pair data for training and validation of our frameworks, limiting our experimental scope. We acknowledge that a more diverse set of real-world datasets, such as video data from cameras mounted on moving manipulator hands or autonomous driving scenarios with vehicle control, would provide a deeper understanding of our proposed frameworks. We aim to highlight the critical importance of simultaneously recording action and video data in dynamic camera scenarios, encouraging future data collection and research methodologies.
>  - **Data size and complexity:** We kept the frame resolution to $64\times 64 \times 3$ due to our computational resource constraints. Video prediction frameworks require significantly higher computational resources to train and evaluate, and we primarily used NVIDIA 3090 systems for our research. With sufficient computational resources, all our frameworks can be trained on high-resolution images to generate more sharper predictions.
>
>  - In our evaluations, we predicted 20 frames into the future based on past 5 frames from the RoAM dataset (recorded at 15fps), effectively predicting the next 1.33 seconds of video from 1/3rd of a second of input. In temporal ablation studies (Fig 5a and 5b), we predicted 15 frames from the last 5 frames, covering a 2-second look-ahead window. Given that mobile robot actuators respond within milliseconds, this 1.3 to 2-second prediction provides sufficient time for obstacle avoidance.
>
>  - Furthermore, for video predictions on partially observable datasets like KITTI, people normally predict the next 20-25 frames from the last 5-10 observed frames (Villegas et al(2019), Gao et al 2022). Thus for our benchmarking we have followed the similar parameters and predicted the next 20 frames.

---

> > ### Author Response · Authors · 2024-12-01
> > **Response 2/2**
> >
> > ### **3. Comparison with State-of-the-Art:**
> >  - **overfitting issues:** The ground truth control values are normalized within the limit of $[0,1]$ to ensure our trained models remain agnostic to actuator limitations across various mobile robot models. The lack of overfitting is evident in Figure 4a, where after timestep 13, the accumulated frame prediction error causes the forward velocity prediction by VG-LeAP to grow exponentially. This highlights the dual-dependent nature of our proposed theoretical frameworks.
> >
> >  - We intentionally maintained very low weights on the loss components related to action error in Equations 6 and 13 to minimize action data overfitting and stabilize overall training. Detailed implementation specifics are provided in the Supplementary material.
> >
> >  - **ablation studies:** We have provided detailed temporal ablation studies on the effect of changing the sampling time between training and test video frames. During testing, we doubled the sampling time-step between frames compared to training (effectively reducing the frame rate so objects appear to move faster) and reported results in Figures 5a and 5b. We will also include a comprehensive ablation study on the effects of different weights for action prediction loss components in Equations 6 and 13, along with the impact of changing latent space size, in our final paper.
> >
> >  - ACPNet: Sarkar et al, Action-conditioned deep visual prediction with roam, a new indoor human motion dataset for autonomous robots. IEEE RO-MAN 2023.
> >
> >
> > We hope this clarifies the key distinctions and contributions of our work. Thank you again for your thoughtful review and questions and we remain open to your feedback.
> >
> > References
> > #### Yang, Shiyuan, et al. "Direct-a-video: Customized video generation with user-directed camera movement and object motion." 2024.
> > #### Wang, Xiang, et al. "Videocomposer: Compositional video synthesis with motion controllability." NeurIPS 36 (2024).
> >
> > #### Villegas et al. High fidelity video prediction with large stochastic recurrent neural networks. In Proceedings of the Thirty-second Advances in Neural Information Processing Systems, NeurIPS 2019
> >
> > #### Gao et al. Li. Simvp: Simpler yet better video prediction. In Proceedings of the IEEE/CVF Conference on Computer Vision and Pattern Recognition (CVPR) 2022

---

### Official Review · Reviewer_XuYh · 2024-11-04

**Soundness:** 2
**Presentation:** 2
**Contribution:** 2
**Rating:** 5
**Confidence:** 3

**Summary:**

The paper proposes three action-conditioned video prediction training frameworks and shows empirical results of their efficacy on a robot video dataset.

**Strengths:**

* The paper adapts action-free VAE and flow-based formulation from video prediction literature to action-conditioned learning.
* Empirical results show benefit of incorporating action information in training in terms of video prediction accuracy.

**Weaknesses:**

* Missing related works. The paper revisits the literature on VAE-based and flow-matching-based video prediction models, without discussing the latter in the section of prior works.
* The method restricts cameras to be static (line 148). This assumption does not hold in general for casual videos outside of the training data being used and limits the applicability of the method.
* The assumption of causality between actions $a_t$ and observed framer $x_t$ is again specific to robot manipulation tasks with fixed cameras as in the RoAM dataset used in this paper. This assumption does not hold in generic videos.
* The paper lacks a discussion of the potential benefit of incorporating actions into video modeling frameworks in general. Action-free video prediction training does not require action annotations and is much more scalable. If the downstream task is robot manipulation, an alternative approach is to train action-free video prediction models and extract robot actions via inverse dynamics [1]. More discussions would help strengthen the paper. If the goal is accurate video prediction itself, then how does the method compare to state-of-the-art video prediction architectures?
* What's the relation of the proposed 3 distinct models? A much more extensive discussion on this would help clarify the motivation of developing three separate frameworks in the paper.
* The paper claims results on incorporating camera motion (line 014) but empirically only evaluate on datasets with fixed cameras.

[1] Learning Universal Policies via Text-Guided Video Generation.

**Questions:**

* How are discussions on diffusion-based video prediction (line 131 - 135) related to camera controls (line 103)?
* What's the action space in RoAM? Is it continuous or discrete? Does the method scale to high-dimensional action space?

---

> ### Author Response · Authors · 2024-11-29
> **Response 1/2**
>
> We appreciate the reviewer's valuable feedback and thoughtful insights. Below are the detailed responses:
> ### **1. Discussion of prior flow-matching and diffusion based models:**
> We have modified the Introduction section in our revised paper to account for the aforementioned clarification while also conforming to the given page limit. The paragraph from Line no 64-73 specifically compares our current work with latent flow based and diffusion based video generation models.
> ### **2. RoAM is a dataset with moving camera:**
> We respectively disagree with the reviewer's assertion that our method considers the camera to be stationary. Our all three models are specifically designed to address video prediction problems in scenarios where the camera is moving. We would further like to clarify that all our 3 proposed models are multi-modal prediction frameworks that learn to predict video as well as the possible future action or camera movement data. In line 148 we described the dimension of the action space of the dynamic platform on which the camera to be mounted like a car or a mobile robot. For example in our case we used the RoAM dataset where the camera is mounted on an autonomously moving mobile robot (Turtlebot3) exploring various indoor settings recording different kinds of human movement and actions like walking, sitting, standing etc. In case of RoAM the camera has forward velocity and turn rate as control actions. Thus in our case the dimension of the action space is 2 and it was normalised and continuous, bounded between [0,1].
>
> We would like to add that our proposed work is built on a completely different premise than that of the robot manipulator task where that camera is mounted on a static platform. We provided further discussion on this in our paper Line 46-63 of our revised paper.
> ### **3. Assumption of Causality between action $a_t$ and image frame $x_t$:**
> We have modified our introduction section in the revised paper from line no 65-93 to clarify our motivation behind our causal model. Our causal model has nothing to do with robot manipulation tasks. To our knowledge Causal-LeAP is the first model that tried to model the causal relationship between the camera motion and the observed image frames in the dynamic camera scenario. We have further derived a novel ELBO loss in Eq 13 for our proposed causal framework. In Causal-LeAP images and actions are modeled as causally interlinked nodes, reflecting the real-world scenario where a robot or vehicle takes an action based on the current observed image and observes the next image state as a consequence.
> ### **4. Benefit of incorporating action in video prediction:**
> We have modified our introduction section to clearly articulate the motivation for incorporating action data in video prediction tasks for moving camera scenarios (lines 46-93). The primary objective of our work is to develop multimodal learning frameworks that simultaneously capture the spatio-temporal dynamics embedded in video data and the camera's motion dynamics through action data.
>
> Our frameworks are specifically designed for dynamic environments such as busy roads or crowded spaces, where autonomous vehicles or robots must navigate and avoid moving obstacles based on real-time camera inputs. This approach fundamentally differs from robot manipulation tasks with pre-computed trajectories [1]. Moreover, our three proposed frameworks uniquely predict both the next camera/robot action and image frame, offering significant potential for advancing motion planning algorithms in complex, uncertain environments with multiple moving obstacles.

---

> ### Author Response · Authors · 2024-11-29
> **Response 2/2**
>
> ### **5  Why 3 distinct models:**
>  We believe our modified introduction section from line 74-93 in the revised paper clarifies the need and distinction between the 3 models VG-LeAP, Causal-LeAP and RAFI.
> ### **6 Relation Between Diffusion-Based Video Prediction and Camera Controls:**
> The discussion from lines 64-73 in our revised paper addresses diffusion-based video prediction models that incorporate camera controls in their generation process. Additionally, the diffusion models mentioned in lines 137-141 (previously 131-135) do not incorporate camera controls in their video generation process.
> ### **7 Action Space of RoAM and  Scalability:**
> The dimension of the action space of RoAM is 2, the forward velocity and the turn rate of the robot on which the camera is mounted. The actions are also continuous and are normalised between the values [0,1] as we wanted our models to be agonistic to the different actuator limitation (max/min forward velocity or turn rate limits) from different kinds of mobile vehicles.
> Given that our frameworks do not assume any specific dimensionality for the camera movement action space, they can scale without significant computational overhead. This scalability is evident in the block diagrams of VG-LeAP and Causal-LeAP in Figures 1a and 2b, which utilize action encoders with dense layers.
>
> In both architectures, we map low-dimensional action spaces to high-dimensional manifolds before concatenating with image features or latent variables. This approach prevents overfitting in the action prediction framework. Similarly, in RAFI, we concatenate action data as additional channels to vectorized image embeddings from VQ-GAN. Consequently, all three proposed models can be readily extended to high-dimensional action spaces.
>
> We hope this clarifies the key distinctions and contributions of our work. Thank you again for your thoughtful review and questions and we remain open to your feedback.
>
> #### [1] Learning Universal Policies via Text-Guided Video Generation.

---

> ### Author Response · Authors · 2024-12-01
> **Gentle Reminder for response to our reply**
>
> Dear reviewer XuYh,
>
> Thank you for your time in providing reviews for our paper. As the final deadline is approaching, we would like to gently remind you that we have submitted our rebuttal addressing the key concerns and questions you raised regarding the RoAM dataset, our motivation for proposing 3 different models, why Causality in video prediction between image and camera action matters  and a detailed discussion of prior action conditioned diffusion and flow based models.
>
> We have provided comprehensive explanations to clarify these points along with a revised version of our paper and look forward to your feedback.
>
> Thank you for your time and consideration.

---

### Official Review · Reviewer_u7qA · 2024-11-05

**Soundness:** 3
**Presentation:** 3
**Contribution:** 2
**Rating:** 3
**Confidence:** 3

**Summary:**

The authors tackle the problem of partial observability video prediction, which deals with video prediction problems in which the camera is also moving, in which case the video is influenced by both the scene dynamics and the camera's motion -- common in autonomous vehicles and robot manipulators.

This work explicitly models camera motion dynamics by extending the observed image state (existing settings) by introducing three models build upon prior works. Two models are based upon SVG-lp that learn image-action priors (LeAP) --
- (i) vg-leap (imagen-action pairs generated using a single stochastic process), and
- (ii) causal-leap (causal relationship between image and action), and
- (iii) RAFI -- that augments the image-action state pair of an existing flow-matching model RIVER.

For learning the latent action prior, two variational approaches are presented -- (i) combined image-action prior derived from both the observed image and action, with the assumption that image and action states are conditionally independent, and (ii) two separate posterior priors learnt for the image and action latent variables assuming causal interlinking.

The experimental evaluations are performed on the RoAM dataset which consists of synchronized image-action pairs recorded with a Turtlebot robot using a stereo camera setup. The dataset has 45 training videos and 5 testing sequences (300k videos sequences of 25 frames each used for training). The models are evaluated on the task of video generation for generating 10 frames conditioned of randomly sampled 5 previous consecutive frames. Perceptual and semantic quantitative metrics are used for evaluation.

**Strengths:**

The problem of partially observable video prediction is quite interesting and has applications in autonomous vehicles that have an onboard camera (such as autonomous cars/taxis), drones (with an onboard camera), and robot manipulators (that have wrist-mounted cameras).

Tackling video prediction under partially observable settings (where the acting agent is not visible on the camera) can be benefit robot applications (for instance pedestrian intent detection and prediction could influence autonomous driver decisions, video prediction networks as world models for robot manipulators etc.)

This work tackles the interesting idea of incorporating (robot) actions (as a learned latent) into the video prediction/generation task that have been traditionally conditioned on just image frames.

The problem statement is interesting the authors motivate it well. The paper is also well-presented and articulated except for a few spelling errors (for instance lossses pg 6, divergance pg 6).

**Weaknesses:**

Although sound, I find the contributions (incorporating actions) to be minimal additions to the existing frameworks. For instance, in VG-leap, the extended image-action state pair is used to condition the SVG-lp model instead of just the images, and the latent posterior approximated with recurrent modules.
Similarly, in Causal-leap, two stochastic posteriors are learned -- one each for image and action, learnt using recurrent modules.
For RAFI, the image latent is concatenated with the action state along the image latent's channel dimension of an existing RIVER model. Given the extensive literature on latent conditioning methods, I would have liked to see comparisons/discussions with different latent conditioning methods for incorporating actions.

I find the experimental evaluations quite weak. The experiments are performed just on the RoAM dataset which has just 45 video sequences for training and 5 for testing/inference I would have liked to see comprehensive comparisons -- for instance on the robot manipulation settings in which case the camera is mounted on the robot's wrist. The Causal-leap model should be evaluated on such a setting -- which has a larger action space (7dof instead of the 2-dimensional in RoAM) that was used to motivate the problem. A similar comparison could have been done on drones for instance.

Evaluations on short sequences. Predicting just 10 frames or evaluating on 25 frame length long video sequences does not quantify as long-term video prediction (which was used to motivate the manuscript). In such short horizon prediction problems, it is harder to quantify the effect of the moving camera. I would have liked to see examples of video predictions that are influenced by external factors (for instance the car turning right due to an obstacle on the left -- collision avoidance being one of the motivations). The quantitative metrics used in the paper evaluate semantic/perceptual quality of the generated videos -- fvd, lpips, vgg-16 etc., and don't necessarily motivate incorporating actions into the video prediction problem. TLDR; how obstacles or other hinderances influence video predictions is quite unclear as the metrics primarily evaluate video quality and not decisions. This would also strengthen the applications of such systems.

**Questions:**

- How does the model perform on long-horizon video prediction problems.
- How is the performance in different robot settings? (for instance, robot manipulators, drones, etc.)
- Were any other latent conditioning methods evaluated?

---

> ### Author Response · Authors · 2024-11-29
> **Response 1/3**
>
> We appreciate the reviewer's valuable feedback and thoughtful insights. Their comments highlighting the significance of our research in mobile robotics, particularly its potential applications in safe navigation and obstacle avoidance, are particularly encouraging. We would like to further emphasize that motion planning in unknown environments with multiple moving obstacles has been a primary motivation behind our Causal-LeAP and VG-LeAP models. We would like to draw the reviewer’s attention to the following points.
>
> ### **1. Novelty:**
> We respectfully disagree with the reviewer’s assertion regarding the lack of novelty in our work. Our contributions represent a substantive departure from current state-of-the-art techniques in understanding spatio-temporal dynamic in video in the presence of camera motion. We have also modified the Introduction section (line 046-093) in our revised paper for clarification while also conforming to the given page limit.
>
> While our VG-LeAP model builds upon the foundational SVG-LP framework—a seminal work in variational video generation, we introduced two theoretical frameworks (i) Conditional Independence and (ii) Causal Dependence, that not only incorporate actions into video prediction but simultaneously predict future actions. Under the assumption of Conditional Independence, we assume that image and action are generated together from a composite latent process. VG-LeAP and RAFI were designed upon this assumption  whereas in case of Cause Dependence in Causal-LeAP, we learn the causal relationship between camera motion and observed image frames using two distinct learned priors. We have provided detailed clarification regarding these two theoretical frameworks in line 74-92.
>
> We have also provided a more clear comparison to other existing video prediction frameworks in partially observable scenarios to our current work in Lines 46-57.
>
> - **Novelty of Causal-LeAP:** To our knowledge Causal-LeAP is the first model that tried to model the causal relationship between the camera motion and the observed image frames in the dynamic camera scenario. We have further derived a novel ELBO loss in Eq 13 for our proposed causal model. The causality assumption and separate latent space model for action prediction hold significant implications for end-to-end image-based reinforcement learning motion planners in robotics, potentially allowing direct integration of the learned action prior into RL control agents.
> ### **2. How obstacles or other hindrances influence video predictions:**
> Our results in Figures 6, 7a, and 7b provide a compelling visualization of our models' performance. Figure 6 displays zoomed samples from Causal-LeAP, VG-LeAP, and SVG-LP alongside ground truth images, showcasing an ego-motion view where the robot encounters a human crossing its path from right to left.
>
> The control plots in Figure 7b reveal nuanced behavioral differences. Upon detecting the human obstacle, Causal-LeAP predicts a clockwise right turn (clockwise being positive here) while maintaining a constant forward velocity. In contrast, VG-LeAP's action model predicts differently—taking an anti-clockwise left turn and continuing to track the human leg in its view (the predicted image frame for VG-LeAP at $t=6$ in figure 6), with a noticeable deceleration in forward velocity as shown in Figure 7a.
> This comparison highlights the distinctive predictive and navigational capabilities of our Causal-LeAP and VG-LeAP models in handling dynamic, obstacle-rich environments. The Gound truth in control action was taken according to the autonomous obstacle avoidance module of RoAM dataset and was based on LiDAR data.
> ### **3. Performance on Long-Horizon Video Prediction**:
> In our evaluations we have predicted 20 frames into the future based on past 5 frames. In case of RoAM the videos were recorded at 15fps. Thus with images from past 1/3rd of a second we predicted the next 1.33 sec of video. In our temporal ablation studies given in Fig 5a and 5b, we have doubled the test sampling time from training, and there we predicted next 15 frames from the last 5 frames. Thus in that case by observing the frames from the last 2/3rd of a sec we are predicting the images for the next 2 sec. Given that the actuators in mobile robots have a response time normally in the order of milliseconds, a look-ahead window of 1.3 to 2 second provides enough response time for avoiding an upcoming obstacle.
>
> Furthermore, for video predictions on partially observable datasets like KITTI, people normally predicts the next 20-25 frames from the last 5-10 observed frames \[1\] Villegas et al(2019),\[2\] Gao et al 2022 \[3\] Sarkar et al 2021. Thus for our benchmarking we have followed the similar parameters and predicted the next 20 frames.

---

> ### Author Response · Authors · 2024-11-29
> **Response 2/3**
>
> ### **4. Comparison with Action Conditioned Latent models:**
> As per the reviewers' suggestions, we have added 2 new paragraphs in our introduction (Lines 57-73 of our revised paper) to discuss the distinction between our proposed frameworks and other prior works  regarding action-conditioned video prediction/generation. Below, the reviewer will find a more detailed version of the same discussion.
>
> The concept of conditioning predicted image frames on action was introduced by [4] Oh et al 2015 in their work on vision-based reinforcement learning, where they conditioned predicted Atari environment images using controller actions. However, these approaches assumed known future actions and lacked a mechanism for predicting or conditioning images when future actions are unknown. [5] Valencia et al. applied the concept of action as an extended system state to Introspective Variational Autoencoder (IntroVAE) architectures. Their approach also shared the same limitation as Oh et al. and assumed complete knowledge of future control actions. Additionally, they learned an image prior distribution that was entirely independent of actions.
>
> Following this same line of research, [6] Ma et al. learnt an action conditioned image prior for object grasping tasks with a manipulator. They benchmarked their framework on an object grasping dataset PandaGrasp and had similarities with the deep visual foresight work by [7] Finn et al and assumed full knowledge of the future end-effector action and state data.  In similar works, [8] Nazari et al. applied manipulator action conditioning to enhance slip predictions for improved gripping, while [9] Nunes et al. established a benchmark for action-conditioned video prediction frameworks to infer manipulator actions from predicted frames. More recently, [10] Petrovich et al.  conditioned a Transformer-VAE architecture with text action tokens to model 3D human motion.
>
> Existing studies, while demonstrating the benefits of incorporating action as an extended state for video prediction, relied on the assumption of known future action data. This assumption may work in controlled environments like stationary robotic manipulators with pre-computed end-effector trajectories, but fails in more dynamic scenarios such as moving cameras in unpredictable environments like busy roads or crowded spaces.
>
> In these complex, stochastic settings, an ideal approach requires the ability to learn and predict platform actions based on past and predicted image frames, and vice versa. Our proposed models, VG-LeAP and Causal-LeAP, directly address this limitation by not only conditioning predicted images on camera action but also predicting the next action based on predicted images.
>
>  ### **5. Performance Across Different Robotic Settings and action spaces:**
>  - **Robotics manipulator and drones:** Currently there are no public dataset available on robotic manipulator actions where the camera is mounted on the moving manipulator itself. The current robotic manipulator dataset BAIR robot push } is configured with a static camera framework recording manipulator actions. Although action data is available due to the stationary camera, this setup bears little relevance to our VG-LeAP or Causal models, which focus on dynamic camera environments with uncertain motion trajectories. Similarly, drone datasets like \[11\] DroneCrowd do not provide synchronized drone actuation data for framework evaluation. We aim for our action-conditioned research to highlight the critical importance of simultaneously recording action and video data in dynamic camera scenarios, thereby encouraging future data collection and research methodologies.
>
>  - **Scaling of Action Space:** Given that our frameworks do not assume any specific dimensionality for the camera movement action space, they can scale without significant computational overhead. This scalability is evident in the block diagrams of VG-LeAP and Causal-LeAP in Figures 1a and 2b, which utilize action encoders with dense layers. In both architectures, we map low-dimensional action spaces to high-dimensional manifolds before concatenating with image features or latent variables. This approach prevents overfitting in the action prediction framework. Similarly, in RAFI, we concatenate action data as additional channels to vectorized image embeddings from VQ-GAN. Consequently, all three proposed models can be readily extended to high-dimensional action spaces.
>
> We hope this clarifies the key distinctions and contributions of our work. Thank you again for your thoughtful review and questions and we remain open to your feedback.

---

> ### Author Response · Authors · 2024-11-29
> **Response 3/3**
>
> References:
>
> #### \[1\] Villegas et al. High fidelity video prediction with large stochastic recurrent neural networks. In Proceedings of the Thirty-second Advances in Neural Information Processing Systems, NeurIPS 2019
>
>
> #### \[2\] Gao et al. Li. Simvp: Simpler yet better video prediction. In Proceedings of the IEEE/CVF Conference on Computer Vision and Pattern Recognition (CVPR)
>
> #### \[3\] Sarkar et al. Decomposing camera and object motion for an improved video sequence prediction. In NeurIPS 2020 Workshop on Pre-registration in Machine Learning
>
> #### \[4\] Oh et al. Action-conditional video prediction using deep networks in atari games. In Proceedings of the Twenty-ninth International Conference on Neural Information Processing Systems, NIPS 2015
>
> #### \[5\] Valencia et al. Action-conditioned frame prediction without discriminator. In International Conference on Machine Learning, Optimization, and Data Science
>
> #### \[6\] Ma et al. Vp-go: A ‘light’actionconditioned visual prediction model for grasping objects. In 2022 International Conference on Advanced Robotics and Mechatronics (ICARM)
>
> #### \[7\] Finn et al. Unsupervised learning for physical interaction through video prediction. In Proceedings of Thirtieth Conference on Neural Information Processing Systems, NIPS 2016,
>
> #### \[8\] Nazari et al. Action conditioned tactile prediction: a case study on slip prediction. 2022.
>
> #### \[9\] Nunes et al. Action-conditioned benchmarking of robotic video prediction models: a comparative study. In 2020 IEEE International Conference on Robotics and Automation (ICRA)
>
> #### \[10\] Petrovich et al. "Action-conditioned 3d human motion synthesis with transformer vae." _Proceedings of the IEEE/CVF International Conference on Computer Vision_. 2021.
>
> #### \[11\] Wen et al. “Detection, Tracking, and Counting Meets Drones in Crowds: A Benchmark”, CVPR 2021

---

> ### Author Response · Authors · 2024-12-01
> **Gentle reminder for feedback to our reply**
>
> Dear reviewer u7qA,
>
> Thank you for your time in providing reviews for our paper. We would like to gently remind you that we have submitted our rebuttal addressing the key concerns you raised regarding the novelty of our work, model performance in different robotic settings, and a detailed discussion of prior latent-conditioned models as the final deadline is approaching.
>
> We have provided comprehensive explanations to clarify these points along with a revised version of our paper and look forward to your feedback.
>
> Thank you for your time and consideration.

---

### Author Response · Authors · 2024-11-25
**Response**

We thank all the reviewers for their valuable feedback. We would also like to add that we have revised our paper based on the feedback provided by the reviewers. We would like to especially draw the reviewers' attention to our revised introduction section from Line 31-93 to provide clarity on the novelty of our work compared to prior works on action-conditioned video generation and the motivation behind our proposed three different models.

Due to a severe illness and sudden hospitalisation, we could not post our replies earlier. Despite the medical interruption, we remain committed to thoroughly addressing the reviewers' valuable feedback and demonstrating the scientific rigor of our research and are open to their further feedback.

---

### Meta-Review · Area_Chair_Gafq · 2024-12-21

**Metareview:**

The paper addresses long-term stochastic video generation with moving cameras, which introduces complex spatio-temporal dynamics and partial observability. Traditional methods focus on pixel-level reconstruction without modeling camera motion. To address this, the authors propose a multi-modal learning framework that incorporates camera actions into the image state. They introduce three models: VG-LeAP: Combines image-action pairs into an augmented state using a single latent stochastic process with variational inference. Causal-LeAP: Establishes a causal link between actions and image frames, learning separate action priors conditioned on image states. RAFI: Integrates image-action states with a conditional flow matching framework, extending to transformer-based architectures.

Strengths:

-- Tackles the important issue of video prediction with moving cameras, applicable to autonomous vehicles and robotics.

-- Effectively integrates camera motion with visual data, extending existing video generation models.

-- Shows performance improvements on the RoAM dataset, highlighting the benefits of action incorporation.

Weakness:

-- The integration of camera motion is incremental and lacks significant innovation compared to recent studies.

-- Evaluations are restricted to the RoAM dataset, limiting the demonstration of generalizability.

-- Does not adequately compare with a wide range of state-of-the-art models or discuss the distinct advantages of the proposed methods.

Despite addressing a relevant problem and showing some empirical benefits, the paper suffers from narrow experimental validation, and inadequate comparative analysis.  After carefully reading the paper, the reviews and rebuttal discussions, the AC agrees with the reviewers on recommending to reject the paper.

**Additional Comments On Reviewer Discussion:**

See the weakness and comments above, there are still remaining concerns from reviewers.

---

### Decision · Program_Chairs · 2025-01-22

Reject